

# Sources of Submicrometre Particles
5
# Near a Major International Airport

**Mauro Masiol[1,2], Roy M. Harrison[1*†],**
**Tuan V. Vu[1], David C.S. Beddows[1]**

**[1] Division of Environmental Health and Risk Management**
**School of Geography, Earth and Environmental Sciences**
**University of Birmingham**
**Edgbaston, Birmingham B15 2TT**
**United Kingdom**

**[2] Center for Air Resources Engineering and Science, Clarkson**
**University, Potsdam, NY 13699, United States**


---

[*] To whom correspondence should be addressed.
Tele: +44 121 414 3494; Fax: +44 121 414 3709; Email: r.m.harrison@bham.ac.uk

[†] Also at: Department of Environmental Sciences / Center of Excellence in Environmental Studies, King Abdulaziz University, PO Box 80203, Jeddah, 21589, Saudi Arabia





**ABSTRACT**
Major airports are often located within or close to large cities; their impacts on the deterioration of
air quality at ground level are amply recognised. The international airport of Heathrow is a major
source of nitrogen oxides in the Greater London area, but its contribution to the levels of
submicrometre particles is unknown, and is the objective of this study. Two sampling campaigns
were carried out during warm and cold seasons at a site close to the airfield (1.2 km). Size spectra
were largely dominated by ultrafine particles: nucleation particles (<30 nm) were found to be ~10
times higher than those commonly measured in urban background environments of London. A set
of chemometric tools was used to discern the pollution arising from aircraft operations and those
from other sources within the city or from the traffic generated by the airport. Five clusters and 6
factors were identified by applying *k*-means cluster analysis and positive matrix factorization (PMF)
respectively to particle number size distributions; their interpretation was based on their modal
structures, wind directionality, diurnal patterns, road and airport traffic volumes and on the
relationship with weather and other air pollutants. Airport emissions, fresh and aged road traffic,
urban accumulation mode and two secondary sources were then identified and apportioned. The
comparison of cluster and PMF analyses allowed extraction of further information. The analysis of
a strong regional nucleation event was also performed to detect its effect upon concentrations. The
fingerprint of Heathrow has a characteristic modal structure peaking at <20 nm and accounts for 30-
35% of total particles in both the seasons. Other main contributors are fresh (24-36%) and aged (16-
21%) road traffic emissions and urban accumulation from London (around 10%). Secondary
sources accounted for less than 6% in number concentrations but for more than 50% in volume
concentration. In 2016, the UK government provisionally approved the construction of a third
runway; therefore the direct and indirect impact of Heathrow on local air quality is expected to
increase unless mitigation strategies are applied successfully.
**Keywords:** Airport; black carbon; size distributions; source apportionment; ultrafine particles





## 1.    INTRODUCTION

Emerging markets, developing economies and globalisation are driving a fast and continuing
growth of civil aviation, which is expected to continue in the next decade (Lee et al., 2009). As a
consequence, the aircraft and road traffic at airports is also increasing, but the information available
on the impact of airport emissions upon air quality at ground level is still inadequate (Webb et al.,
2008; Masiol and Harrison, 2014). The quantification of airport impacts on local air quality is
complicated by the complexity of multiple mobile and static emission sources, with many airports
being located near to major cities, highways or industrial plants. Under this scenario, the
development of successful strategies for emission mitigation and the implementation of measures
for air quality control to meet regulatory standards require an exhaustive quantification of the
contribution of airport emissions to the total air pollution load.

London Heathrow (LHR) is one of the world's busiest international airports: it is ranked 1st in
Europe for total passenger traffic (ACI, 2016). Its role in driving the economic affluence and vitality
of the Southern UK is indisputable: it accommodates more than 1250 flights every day and serves a
total of 72.3 million passengers year$^{-1}$. LHR is composed of 5 terminals and 2 runways: northern
(3.9 km-long) and southern (3.7 km). Currently, runways operate near their maximum capacity,
with a consequent increase in the potential for delays when flights are disrupted. Since 2007, the
proposal for expanding LHR with a 3rd runway and a 6th terminal has been intensely debated in
UK. The main reasons supporting its expansion are: (i) the expected increase of resilience to
disruption caused by congested flight traffic; (ii) the improvement of its connectivity with a
profitable network of both direct long haul air routes and national flight connections; (iii) the
potential to directly enhance the economic growth of the London area. On the contrary, opposition
to LHR expansion highlights the potential increases in air pollution and noise, the community
destruction and argues in favour of alternative options with fewer local impacts, such as the
improvement of other airports in the southern UK or the building of a new airport in the Thames

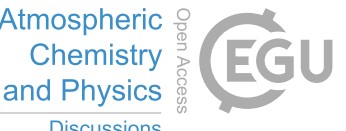

Estuary (East of London). Despite this, in 2016 the UK government provisionally approved the
construction of a third runway.

Greater London is one of the few UK locations not fully achieving the EU and national air quality
standards: in 2015 nitrogen dioxide breached the hourly and annual limit values for health, while
ozone exceeded the long-term objective (DEFRA, 2016). On the other hand, the mass concentration
of particulate matter (PM), which is the standard current metric for measuring and controlling the
exposure to airborne particles, was fully met for both $PM_{10}$ and $PM_{2.5}$. However, it has been widely
demonstrated that even PM mass concentrations below guidelines and standards set by legislatures
or international organizations may increase acute and chronic effects and mortality (e.g., Shi et al.,
2015). In this situation, the use of mass concentration as a sole metric for measuring the levels or
airborne particles has the disadvantage of taking greatest account of accumulation and coarse mode
particles, which account for most of the mass. Consequently, the impact of the finest particles is not
accounted for directly. This issue raises serious questions for the air quality standards: biological
evidence associates the exposure to ultrafine particles (UFPs, <100 nm) with adverse effects upon
human health (e.g., Knibbs et al., 2011; Strak et al., 2012; Ostro et al., 2015; Lanzinger et al., 2016).
At the current time, there is still limited knowledge of what specific characteristic or association of
characteristics may dominate the particle toxicity, and the consequent  health outcomes (Atkinson et
al., 2010; Strak et al., 2012, Vu et al., 2015a); nevertheless it is well recognised that  UFPs can
reach the deepest regions of the lung (Salma et al., 2015) and may have orders of magnitude higher
surface area to mass ratios compared to larger particles.  They offer more surface for the absorption
of volatile and semi-volatile species (Kelly and Fussell, 2012; Strak et al., 2012).  However, there
are currently no ambient air quality standards or guidelines to drive the regulation of UPF.

The goal of this study was to investigate the impacts of a major airport (LHR) serving a megacity
(London) upon the levels of submicrometre particles and equivalent black carbon (eBC) and to





apportion those impacts to aircraft, road traffic and other sources typical of large cities with
airports. This task was performed by collecting air quality data at a site downwind of LHR and by
applying a series of chemometric tools. The potential sources of submicron particle number
concentrations (PNC) are investigated by applying two source apportionment methods: cluster
analysis and positive matrix factorisation (PMF). Thus, the origin of the airport plumes was
spatially assessed by matching results with local meteorological data, air mass movements, levels of
common air pollutants, $PM_{2.5}$ mass concentration and its chemical speciation as an indicators of
source location and formation mechanisms. Finally, the disaggregated source profiles are used to
trace the factors affecting the pollutant levels, such as atmospheric dispersion and processing of
aircraft emissions as well as of road traffic.

This study was carried out under the Marie Skłodowska-Curie project CHEERS (Chemical and
Physical Properties and Source Apportionment of Airport Emissions in the context of European Air
Quality Directives, call: FP7-PEOPLE-2012-IEF, project no. 328542).

**2.        MATERIALS AND METHODS**
**2.1    Study Area and Dates**
The summer (warm season) campaign took place from 13 August to 12 September 2014 and the
winter (cold season) campaign from 19 December 2014 to 20 January 2015.  The Greater London
area hosts more than 8.5 million inhabitants and LHR is located west of London (Figure 1).
Consequently, air quality in the surroundings of the airport may be affected by the advection of air
masses from the city, with the associated high levels of pollutants emitted from traffic, energy
demand for domestic heating and local industries. Airport activities may also contribute to air
pollution advected to the city when LHR is upwind, with consequent potential impacts upon public
health. In addition, as LHR attracts a large number of passengers and workers, the emissions from
large volumes of road traffic generated by the airport and the nearby M4 and M25 motorways are





difficult to discriminate from non-airport-related road traffic. Due to this complex scenario, the
contribution of LHR is difficult to differentiate from the urban background pollution, as already
reported by previous modelling and experimental studies (Farias and ApSimon, 2006; Masiol and
Harrison, 2015).

Various studies have attempted to quantify the effect of LHR upon air quality, mainly focusing on
the nitrogen oxides ($NO_x=NO+NO_2$), which are well-known tracers for aircraft engine exhausts
(e.g., Herndon et al., 2008; Masiol and Harrison, 2014 and references therein), but also arise from
other combustion sources. For example, Carslaw et al. (2006) estimated that airport operations in
2001/4 accounted for ~27% of the annual mean $NO_x$ and $NO_2$ at the airfield boundary and less than
15% (<10 µg m$^{-3}$) at background locations 2-3 km downwind of the airport. Similar results were
found for the 2008/9 period using model evaluation (AEA, 2010) and for the 2005/12 period using
experimental data analysis (Masiol and Harrison, 2015). This latter study also reported that PM
mass concentrations at eight sites all around LHR were always well below the EU and UK limit.

**2.2**      **Site Description**
Two intensive sampling campaigns (each 1 month-long) were carried out during warm (August-
September 2014) and cold (December 2014-January 2015) periods at Harlington (Figure 1). Data
from the site are quality assured as part of the UK Automatic Urban and Rural Network under the
auspices of the UK Department for Environment, Food and Rural Affairs (DEFRA; http://uk-
air.defra.gov.uk/) and the site was selected as well located to sample the plumes from the airport
emissions. The site lies 1.2 km N of the northern runway and is located inside a playground, close
to a secondary road and near the village of Harlington. This is the location selected for the
construction of the 3rd runway. The site is categorised as "urban industrial" by DEFRA and it is
therefore more indicative of community exposure rather than direct fresh aircraft emissions.
Consequently, it is a good point to quantify the particles generated by the airport after a relatively





short ageing and dispersion in the atmosphere, and is more indicative of the fingerprint of aircraft
emissions affecting communities than data collected alongside the runway or in the airport apron
areas. In addition, previous studies have reported that the site is strongly affected by the plume from
the airport (Carslaw et al., 2006; Masiol and Harrison, 2015). Prevailing winds from the 3rd and 4th
quadrants are recorded in both summer and winter (Figure SI1): under such circulation regimes,
Harlington lies just downwind of LHR. However, the site is also affected by pollutants arising from
the large volumes of road traffic generated by the airport: Tunnel Rd., the main access to LHR from
the M4 motorway lies 800 m west, as well as the nearby M4 (640 m north) and M25 (~3.5 km east)
motorways, major roads (Bath Rd, part of A4, passes 900 m south; A30 lies 2.8 km SE).  The
village of Harlington (~400 m west) and the conurbation of London are other potential external
sources.

## 172    2.3    Instrumentation Suite

Ultrafine particle counts and their size distributions from 14.3 to 673.2 nm were measured at 5 min
time resolution using a SMPS (scanning mobility particle sizer spectrometer) comprising a
electrostatic classifier TSI 3080 with a long differential mobility analyser (TSI 3081) and a CPC
(condensation particle counter, TSI 3775) based on condensation of *n*-butyl alcohol (Fisher
Scientific, ACS). The SMPS operated at a sheath air to aerosol flow ratio of 10:1 (sheath and
sample air flow rates were 3.0 and 0.3 L min$^{-1}$ respectively, voltage 10-9591 V; density 1.2 g/cc;
scan time 120 s, retrace 15 s;  number of scan 2) while the CPC operated at low flow rate (0.3 L
min$^{-1}$). The use of 5 min resolved spectra has already been used successfully for source
apportionment purposes at an airport (Masiol et al., 2016).

eBC was also measured at 5 min resolution using a 7-wavelength aethalometer (Magee Scientific
AE31). The aethalometer operated with an inlet cut-off head to collect PM with aerodynamic



diameter of <2.5 μm ($PM_{2.5}$). eBC was derived from the absorbance at 880 nm wavelength (Petzold
et al., 2013).

Instruments were installed into a plastic/metal case designed for sampling purposes: (i) air inlets
were ~1.8 m over the ground and were composed of conductive materials to avoid particle losses
and sampling artefacts; (ii) the case was cooled by fans in summer and was warmed by an electrical
tubular heater in winter for maintaining an indoor air temperature within an acceptable range for
running the equipment (temperature inside the case was recorded and periodically checked); (iii)
instruments were isolated from vibration using rubber pads and foam foils. Devices were fully
serviced, calibrated by authorised companies and underwent internal cross-calibrations with other
similar instruments under lab conditions. Moreover, frequent periodic checks, maintenance of
instruments and cleaning of inlets was performed throughout the sampling campaign.

Classical air pollutants (NO, $NO_2$, $NO_x$, $O_3$, $PM_{10}$, $PM_{2.5}$) were measured at Harlington with 1 h
time resolution. Gaseous species were analysed using automatic instruments according to European
standards and National protocols: EN 14211:2012 for nitrogen oxides and EN 14625:2012 for
ozone. $PM_{10}$ and $PM_{2.5}$ were analysed using tapered element oscillating microbalance and filter
dynamics measurement system (TEOM-FDMS) to provide measurements accounting for volatile
($VPM_{10}$, $VPM_{2.5}$) and non-volatile ($NVPM_{10}$, $NVPM_{2.5}$) fractions. Quality assurance and quality
control procedures followed the standards applied for the Automatic Urban and Rural Network
(AURN) and the London Air Quality Network (LAQN). Instruments were routinely calibrated, and
every six months were fully serviced and underwent intercalibration audits.

Weather data were measured hourly by the Met Office at LHR; met data include wind direction and
speed, atmospheric pressure, air temperature, relative humidity (RH), visibility, rain and solar
irradiance.




During the two campaigns, 24-h $PM_{2.5}$ samples were also collected on quartz filters using a high
volume air sampler (TE-6070, Tisch Environmental, Inc.) and analysed for the daily concentrations
of major $PM_{2.5}$ components: organic carbon (OC) and elemental carbon (EC) by thermo-optical
analysis (EUSAAR_2 protocol) and major inorganic ions ($Na^+$, $K^+$, ammonium, nitrate, sulphate,
oxalate) by ion chromatography. Analytical methods are reported in detail in Yin et al. (2010). The
results of the chemical speciation of $PM_{2.5}$ are presented in a companion paper (in preparation) and
are used in this study only to assist the interpretation of PMF results.

**2.4       Data Handling and Chemometric Approaches**
Data were analysed using R version 3.3.1 (R Core Team, 2015) and a series of supplementary
packages, including 'Openair' (Carslaw and Ropkins, 2012). Preliminary data handling and clean-
up were carried out to check the robustness of the dataset, detect anomalous records and to delete
extreme outliers. SMPS data with unreliable behaviour or instrument errors were completely
deleted. All remaining data are used for descriptive statistics, but data greater than the 99.5th
percentile were further removed for explorative, cluster and PMF analyses. Missing data for other
variables were linearly interpolated between the nearest values of the time series.

The particle number size distributions (PNSDs) were firstly grouped by applying a $k$-means cluster
analysis. The full method is exhaustively discussed in Beddows et al. (2009; 2014) and aims to
assemble single spectra into $k$ clusters. The clustering groups observations with spectra similar to
their cluster centroids (means), i.e. observations that are likely generated by the same set of
formation processes or emission sources. The optimum number of clusters ($k$) was determined by an
optimisation algorithm based on the spectral shapes (Beddows et al., 2009). The choice to apply $k$-
mean clustering method was based on several reasons: (i) Salimi et al. (2014) reported that $k$-means
is the best performing clustering among others methods tested on PNSD data; (ii) $k$-means is a well-



established method which has been largely applied over a number of different sites (e.g., Dall'Osto
et al., 2012; Wegner et al., 2012; Beddows et al., 2014; Brines et al., 2014; 2015); and (iii) the
method was previously applied successfully to airport data (Masiol et al., 2016).
PMF analysis was performed by applying the USEPA PMF5 model. Details of the PMF model are
reported elsewhere (Paatero and Tapper, 1994; Paatero, 1997; USEPA, 2014), while the best
practice and standards are extensively reviewed in several papers (e.g., Reff et al., 2007; Belis et al.,
2014; Brown et al., 2015; Hopke, 2016). SMPS data at 5 min resolution were used as the PMF input
matrix. Uncertainties associated with SMPS data were estimated according to the empirical method
proposed by Ogulei et al. (2007). Uncertainty for the total variable (total particle number
concentration, PNC) was set at 300% of the PNC concentration and also marked as "weak" to avoid
it driving the profiles.

A series of additional tools were used to analyse the raw data, link source apportionment results to
other variables, such as local atmospheric circulation and regional/transboundary transport of air
masses. Briefly, polar plots aim to map pollutant average concentrations by wind speed and
direction as continuous surfaces (Carslaw et al., 2006), while polar annuli plot by wind direction
and hours of the day. The potential locations of distant sources were assessed using back-trajectory
analysis and a concentration weighted trajectory (CWT) model (Stohl, 1998). Back-trajectories
were computed with the HYSPLIT4 model (Stein et al., 2015; Rolph, 2016) using NCEP/NCAR
reanalysis gridded meteorological data. Set-up: -96 h with a starting height of 500 m a.g.l. CWT is a
method of weighting trajectories with associated concentrations to detect the most probable source
areas of long-range transports of pollutants; it has been used and reviewed in a number of prior
studies (e.g., Stohl, 1996; Lupu and Maenhaut, 2002; Squizzato and Masiol, 2015).





## 3. RESULTS AND DISCUSSION

### 3.1 Overview of Data

The wind roses during the two sampling periods are provided in Figure 1. Descriptive statistics of

all collected variables are aggregated as boxplots in Figure 2a. Some additional variables are also

computed to help the interpretation of results. The $NO_2/NO_x$ ratio is indicative of the partitioning of

nitrogen oxides, while the levels of oxidants ($OX=O_3+NO_2$, expressed in ppbv) can be used to

roughly assess the oxidative potential in the atmosphere (Kley et al., 1999; Clapp and Jenkin, 2001).

These two new variables are useful in investigating the atmospheric chemistry behind the NO-$NO_2$-

$O_3$ system. Delta-C (the difference between absorbance at 378 and 880 nm, also called UVPM) was

also computed. This variable was largely used as a proxy to estimate the fraction of carbonaceous

material emitted by biomass burning (e.g., Sandradewi et al., 2008; Wang et al., 2011). However,

Delta-C results should be used with caution: Harrison et al. (2013) showed that there are probably

other UV absorbing contributors than wood-smoke to the aethalometer signal. This way, Delta-C is

used here only for qualitative purposes.

PNSDs were initially split into 3 ranges: nucleation (14-30 nm), Aitken nuclei (30- 100 nm) and

accumulation (>100 nm). On average the total PNC during the warm season was $1.9 \cdot 10^4$ particles

$cm^{-3}$, of which $1.1 \times 10^4$, $6.4 \times 10^3$ and $1.5 \times 10^3$ particles $cm^{-3}$ were classified as nucleation, Aitken

and accumulation ranges, respectively. During the cold season, the total average PNC was $2.2 \times 10^4$

particles $cm^{-3}$, composed of $1.4 \times 10^4$, $6.3 \times 10^3$ and $1.4 \times 10^3$ particles $cm^{-3}$ as nucleation, Aitken

and accumulation ranges, respectively. Concentrations lie between those of London, Marylebone

Road (kerbside) and London, North Kensington (background), and nucleation particles were ~10

times higher than the annual average measured in North Kensington as reported by Vu et al. (2016),

while Aitken particles were 1.9 times higher. It is therefore evident that the main difference lies in

the concentration of the finest size ranges: in both seasons, spectra were dominated by UFP

($D_p<100$ nm) particles (~92% of total PNC), which only accounted for ~12% of total particle



volume concentration (PVC, computed by approximation to spherical particles). On the other hand,
accumulation mode particles accounted for ~8% of PNC and ~88% of PVC volume.

The high levels of total PNC are not surprising. Several studies have reported large increases in
PNC near airports. For example, Hsu et al. (2013) and Stafoggia et al. (2016) detected substantial
increases of PNC values at the airports of Los Angeles (CA, USA) and Rome Ciampino (Italy),
respectively, in the few minutes after take-offs, especially downwind, while landings made only a
modest contribution to ground-level PNC observations. Hsu et al. (2014) observed that departures
and arrivals on a major runway of Green International Airport (Warwick, RI, USA) had a
significant influence on UFP concentrations in a neighborhood proximate to the end of the runway.
In a study carried out at the Los Angeles international airport (CA, USA), Hudda et al. (2014)
concluded that emissions from the airport increase PNC by 4- to 5-fold at 8–10 km downwind of
the airfield, while Shirmohammadi et al. (2017) reported that the daily contributions of the airport
to PNC were approximately 11 times greater than those from three surrounding freeways. Hudda et
al. (2016) reported that average PNC were 2- and 1.33-fold higher at sites 4 and 7.3 km from the
Boston (MA, USA) airport when winds were from the direction of the airfield compared to other
directions. The site used in this study is even closer to the airfield (1.2 km) and is also affected by
strong non-airport sources, such as road traffic emissions due to the presence of two motorways and
several busy roads (frequently congested).

During the warm season, the average concentrations for other pollutants followed the order (in µg
m$^{-3}$): $NO_x$ (49)> $O_3$ (31)> $NO_2$ (31)> $PM_{10}$ (20)> $NVPM_{10}$ (16)> $PM_{2.5}$ (14)> NO (12)> $NVPM_{2.5}$
(11)> $VPM_{10}$ (4)> $VPM_{2.5}$ (3.2)> eBC (2.4)> Delta-C (<0.1). The average concentrations during the
cold season were: $NO_x$ (83)> $NO_2$ (38)> $O_3$ (34)> NO (29)> $PM_{10}$ (18)> $NVPM_{10}$ (14)> $PM_{2.5}$
(13)> $NVPM_{2.5}$ (9.8)> $VPM_{10}$ (4.3)> $VPM_{2.5}$ (3.4)> eBC (2.1)> Delta-C (0.2). These values are
similar to the average concentrations for common air pollutants measured in the vicinity of LHR


reported by Masiol and Harrison (2015) over an 8 year period (2005-2012). Consequently, despite
the intensive sampling campaign carried out in this study, results may be considered representative
of the average levels of air pollution recorded at Harlington.
Since the data were generally not distributed normally, the nonparametric Kruskal-Wallis one-way
analysis of variance was used to test the difference of concentrations over the two periods: almost
all variables are different at the 0.05 significance level, except NO, $NO_x$ and $O_3$. This result
indicates a seasonal effect upon air quality in the LHR area and suggests investigating the sources
over the two periods separately.

The PNSDs are shown in Figure 3. Spectra are categorised by time of day (7am-7pm and 7pm- 7am
local time). In addition, the particle volume size distributions (PVSDs) are also provided. Results
show that in both seasons the nocturnal data are shifted toward coarser modes with respect to the
diurnal mean PNSD, while the modal structure of PNVDs is almost constant throughout the day.

The diurnal cycles of most important variables are shown in Figure 2b. Generally, diurnal cycles
derive from the interplay of emissions, dispersion and atmospheric chemical processes.
Consequently, they need to be investigated along with patterns for airport and motorway traffic
(Figure 2b and Figure SI2, respectively), and as polar annuli (Figures SI3 and SI4) and polar plots
(Figures SI5 and SI5), which give preliminary insights upon the origin and spatial location of most
probable emission sources. Airport traffic undergoes to some restrictions to limit noise community
disturbance: flights are generally constant from 6 am to 8 pm and are kept at minimum overnight,
with no departures normally scheduled between 11 pm and 6 am (Figure 2b). Road traffic is more
difficult to define. Data for M4 and M25 motorways are provided by the UK Department for
Transport: data for the M4 motorway show typical morning (7-8 am) and evening (5-6 pm) peaks
due to rush hours, but this pattern is not well-resolved for the M25 (Figure SI2). In addition, despite
it being likely that traffic on minor and local roads also follows patterns dominated by rush hours,



traffic generated by the airport is more difficult to characterise, with Tunnel Rd. and other busy
roads serving LHR being frequently congested.

Nucleation particles are likely associated with aircraft movements: the daily pattern shows almost
constant concentrations between 7 am and 10 pm, while levels drop to near-zero overnight; the
maximum average concentrations are recorded for winds blowing from the SW quadrant, i.e. the
airfield and, in particular, the location of the main LHR terminals.  As a consequence of the
dominance of nucleation particles over size spectra, also total PNC follows this pattern. On the
contrary, accumulation particles appear to be associated with road traffic, i.e. daily cycles show
typical rush hour peaks and increases for winds blowing from northern sectors. Aitken nuclei
exhibit an intermediate behaviour between nucleation and accumulation particles: two different
patterns can be found, which are more consistent with road traffic in summer and with aircraft
traffic in winter.

Nitrogen oxides are key air pollutants for this study: (i) $NO_2$ levels do not fully fulfil the air quality
assessment Limit Values for health (1 h and annual mean) in the Greater London urban area
(DEFRA, 2016); (ii) they can be good tracers for airport emissions, since $NO_2$ is the main species of
nitrogen oxides emitted by turbofan engines at idle, while NO is the dominant species at higher
thrust (Wormhoudt et al., 2007; Masiol and Harrison, 2014); (iii) they are also emitted from road
traffic mainly as NO, although recent non-attainments of $NO_2$ standards in Europe have been linked
to the growing proportion of diesel-powered vehicles, which have higher primary (direct) emissions
of $NO_2$ (Carslaw et al., 2007; Grice et al., 2009; Anttila et al., 2011; Cyrys et al., 2012). In addition,
nitrogen oxides and atmospheric oxidants are strongly linked by a series of chemical reactions
which are responsible for their partitioning between NO and $NO_2$ (Finlayson-Pitts and Pitts, 2000;
Seinfeld and Pandis, 2006). To date, $NO_x$ has been thoroughly investigated at LHR (Carslaw et al.,
2006; Masiol and Harrison, 2015): it was estimated that the upper limit contribution of LHR



activities to $NO_2$ at Harlington during the 2001-2012 period was ~15-17%, while that for NO was
~10%. In this study, nitrogen dioxide exhibits two typical rush hour peaks, as previously also
observed at the London, North Kensington urban background site (Bigi and Harrison, 2010), and its
concentration increases for winds blowing from all quadrants, suggesting a mix of different sources,
including airport, road traffic and other combustion emissions. Nitric oxide only shows the morning
rush hour peak and northern directionality (toward the M4 motorway) in summer, while in winter it
lacks any significant pattern. The difference between the patterns of NO and $NO_2$ during the two
periods is also confirmed by the $NO_2/NO_x$ ratio, which shows a morning rush hour minimum in
summer as a consequence of fresh NO emissions, while it is less variable in winter (Figure 2b).

In 2015, ozone met the EU target value, but not the long-term objective in the Greater London area
(DEFRA, 2016). In this study, it does not present any wind directionality and exhibits an evident
daily peak in the mid-afternoon, i.e. when the photochemical activity is enhanced by the higher
solar irradiation and the boundary layer depth is greatest, while a second peak in the early morning
corresponds to a minimum in NO (Figure 2b).

Despite some studies indicating that airports are strong sources of black carbon (Dodson et al.,
2009), other studies report no strong relationships with the flight activity (Masiol et al., 2016; Hsu
et al.,2016). Similarly to $NO_2$, aethalometer data also shows typical patterns of road traffic-
influenced sites for all wavelengths, with two daily peaks corresponding to the hours with higher
traffic. However, Delta-C does not present any evident pattern. eBC shows increased concentrations
when winds blow from northern sectors (plus SE in winter); which excludes airport activities as
being a dominant source in the study area.

Particulate matter ($PM_{10}$ and $PM_{2.5}$) has very weak diurnal patterns. Its wind directionality shows
evident increases for northerly winds. It is therefore evident that PM mass concentrations are



dominated by non-airport sources, i.e. regional secondary pollutants, traffic from the nearby M4 or
background pollution from London. $PM_{2.5}$ concentrations normally do not exceed the Limit Values
in the Greater London area (DEFRA, 2016).

**3.2**    ***k*-means Cluster Analysis**
The clustering algorithm extracted 5 clusters for both periods. The number of clusters was selected
according to the optimisation algorithm, i.e. local maxima in the Dunn indices and silhouette
(Beddows et al., 2009). The extraction of 5 clusters represents a good compromise for the
interpretation of spectral observations. Hussein et al. (2014) reported that is not prudent to describe
the spectra with few clusters (2-4), which are not sufficient to explain variations and detailed
differences in the PNSD observed in the urban atmosphere. On the other hand, they also reported
that extracting too many (>10) clusters may make the aerosol source attribution more challenging.

The cluster centroids (mean spectra of each cluster), the 10th, 25th, 75th and 90th percentile, the
hourly counts patterns and resulting wind roses are shown in Figure 4 and 5 for the warm and cold
season campaigns, respectively. Despite extracted clusters exhibiting significantly different modal
structures for PNC, no differences can be observed for the particle volume size spectra, which all
show a unimodal peak at approx. 200-300 nm.

*3.2.1*    *Warm season*
During the warm season, 20% of total clustered observations were grouped in *cluster 1*. It presents a
sharp peak for nucleation particles which extends below the SMPS detection limit (14 nm), a large
increase in frequency during the afternoon hours (noon to 7pm) and its wind rose shows that this
spectrum shape mostly occurs when the prevailing wind blows from SW. Aircraft are known to
emit particles in the nucleation range (e.g. Mazaheri et al., 2009;2013; Masiol and Harrison, 2014;
and references therein; Lobo et al., 2015) and the wind rose is also compatible with an origin from

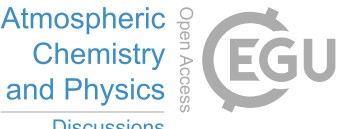

the airfield and the main LHR terminals. However, a similar PNSD profile and a similar daily
pattern was also reported in North Kensington (London background) by Vu et al. (2016) and was
associated with nucleation events. Its interpretation can thus be associated either with airport
activities or photochemical nucleation.

*Clusters 2 and 3* account for 19% and 23% of observations, respectively. While cluster 2 shows a
main peak in number concentrations at 30-40 nm, cluster 3 is bimodal (14 and 60-70 nm). Both
clusters exhibit similar hourly count profiles with most of the counts occurring overnight. This
pattern is largely attributable to the dynamics of the mixing layer, since the diurnal cycles are the
mirror image of the ambient air temperature (Figure 2b). Because of this, both clusters are strongly
affected by the reduced height of the mixing layer occurring overnight. In addition, the role of the
nighttime nitrate formation through condensation of $NH_4NO_3$ and the heterogeneous reactions of
$N_2O_5$ and $NO_3$ on pre-existing particles cannot be ignored (Seinfeld and Pandis, 2006; Bertram and
Thornton, 2009; Brown and Stutz, 2012). However, such clusters occur under different wind
regimes, as the wind roses indicate two different potential source locations: cluster 2 shows a
possible origin from W sectors, while cluster 3 indicates the NE. From this we can infer that cluster
2 likely represents PNSD shaped by: (i) regional aerosols, since the wind directionality suggests an
origin from regions west of London, an area with a lower density of anthropogenic sources, and (ii)
emissions from the M25 motorway and Tunnel Road, i.e. it can be influenced by aged road traffic
emissions. This latter interpretation is also supported by the presence of a peak in the hourly counts
corresponding to the morning rush hours. On the other hand, cluster 3 likely represents the particle
size spectra mainly shaped by primary and secondary aerosols advected from the most urbanised
areas, i.e. it can be likely associated to the urban background of London.

The last two clusters are probably associated with road traffic: vehicle exhaust emissions peak in
the Aitken and accumulation modes with the size ranging from 20 nm to 500 nm (Vu et al., 2015b,


and references therein). *Cluster 5* accounts for 14% of observations and reveals a unimodal
structure peaking at 25 nm. The hourly count pattern exhibits two maxima (6-8 am and 4-8 pm)
related to morning and evening rush hours. The wind rose shows that observations in this cluster
mostly occur when air masses blow from westerly sectors, which are compatible with the location
of motorways and Tunnel Rd, the main roadway linking LHR to the M4 motorway. In addition, it
can be noted that the wind rose exhibits high percentages of high speed winds from W. This pattern
is compatible with fresh road traffic emissions.

*Cluster 4* represents 25% of total observations. It peaks at smaller particle sizes, but also shows a
wide hump at 50-150 nm. It is recognised that road traffic contributes to a large range (30-200 nm)
of PNSD in the urban atmosphere (e.g., Yue et al., 2008; Costabile et al., 2009; Harrison et al.,
2011), which is compatible with this cluster spectrum. In addition, the hour count profile presents a
huge maximum during daytime with possibly 3 maxima (morning and evening rush hours plus mid-
afternoon); this pattern is the mirror image of those for clusters 2 and 3. The directional analysis
shows increased levels when air masses move from the sectors more affected by traffic: London
(NE), M4 (N) and M25 (W) motorways and Tunnel Rd (W). It may represent the typical spectra
recorded during daytime and can be associated with aged anthropogenic emissions, mostly due to
road traffic.

*3.2.2    Cold season*
Unfortunately, the atmospheric circulation during the cold season mostly experienced winds
blowing from the SW quadrant, while the NE sectors were poorly represented. As a consequence,
the limited extent of the wind directionality analysis may blur the interpretation of results. In
addition, the limited number of observations for air pollution advected from the Greater London
area may have affected the detection of the urban background.





*Clusters 1 and 5* account for 24% and 17% of total observations, respectively. They occur under
comparable wind regimes (from SW) and timing (increased counts during daytime). While the
diurnal pattern of cluster 1 has the same shape as the LHR aircraft movement profiles (Figure 2),
cluster 5 is more comparable with cluster 1 for the warm season (maximum in the early afternoon).
However, their spectra are quite different: cluster 1 has a main mode at 20-25 nm, while cluster 5
peaks at 15 nm. Based on the prevailing wind directionality, they can both be linked to airport
activities. A close analysis of wind roses reveals that cluster 5 occurs at significantly higher wind
speed regimes than cluster 1 (Mann-Whitney-Wilcoxon test at 0.05 significance level), i.e. average
wind speeds of 8.3 and 5.9 m s$^{-1}$, respectively. A possible interpretation is that cluster 5 represents
fresher airport emissions (this may also explain the high similarity with the cluster 1 for the warm
season), while cluster 1 depicts the airport emissions which have undergone more aging. The aging
of freshly emitted particles in the atmosphere may involve condensation, evaporation and
agglomeration processes and has been demonstrated to be a major mechanism in altering aerosol
PNSD (e.g., Shi et al., 1999; Kim et al., 2004; Zhang et al., 2005; Zhou et al., 2005; Zhang et al.,
2011; Harrison et al., 2016); this effect was also observed for particles emitted by road traffic in
London (Dall'Osto et al., 2011). Another possible interpretation is that one cluster could represent
the PNSD mainly influenced by aircraft engine emissions, while the other is related to other on-
airport sources, e.g., airport ground service equipment, emissions from auxiliary power units (small
on-board gas-turbine engines) or ground power units provided by the airport. However, this latter
interpretation is less probable, since the spatial extent and temporal pattern of these two sources is
the same (airfield) and, thus, they are expected to be much better mixed.

*Cluster 2* (16% of observations) extends over a wide size range (20 to 150 nm) and presents a daily
pattern likely attributable to the dynamics of the mixing layer (the pattern is the mirror image of the
ambient air temperature). In winter, there is a stronger effect of the mixing layer dynamics on the
air quality due to the presence of more frequent low level thermal inversions which may build up





the pollutants at ground-level especially overnight. Consequently, this cluster cannot be linked to
any specific primary anthropogenic source in the study area, and is likely representative of spectra
mostly shaped by the drop of the mixing layer height and the formation of secondary aerosols.

*Cluster 3* accounts for 20% of data during the cold season. The size spectrum, the wind rose and,
partially, the hourly count profile well relates to cluster 5 for the warm season (attributed to fresh
road traffic emissions). However, the diurnal pattern also presents a high number of counts at 3-5
am, i.e. not compatible with rush hours. Wood smoke is recognised to peak around 100 nm (e.g.,
Chandrasekaran et al., 2013; Vu et al., 2015b). A possible interpretation is that observations
included in this cluster may represent PNSDs dominated by both traffic but influenced by domestic
biomass combustion.

*Cluster 4* (22%) peaks at 17 nm and also shows a wide hump at 50-150 nm. Its diurnal pattern
shows a marked maximum occurring on the afternoon and is mostly represented under westerly
winds regimes. Considering the differences between the two campaigns, it has similar
characteristics to cluster 4 for the warm season. Thus, it can be interpreted as typical of spectra
recorded during daytime and associated with the aging of anthropogenic emissions, mostly due to
road traffic.

**3.3     PMF Analysis**
The best PMF solutions were identified: (i) by investigating solutions between 3 and 10 factors; (ii)
by considering the minimization of the function $Q$ with respect to the expected (theoretical) value
and its stability over multiple (n=100) runs, (iii) by obtaining low values for the sum of the squares
of the differences in scaled residuals for each base run pair by species; (iv) by minimizing the
number of absolute scaled residuals over ±3 and by keeping them symmetrically distributed; (v) by
keeping the result uncertainties calculated by bootstrap (BS, n=200) and displacement (DISP)





methods within an acceptable range (Paatero et al., 2014); (vi) by obtaining modelled total variable
(PNC) successfully predicted ($R^2 > 0.9$ and slopes ≈1); and (vii) by avoiding the presence of edges
in the G-space plots (Paatero et al., 2002) and, then, the presence of hidden/unresolved sources.

The interpretation of PMF results was then attempted by considering: (i) the knowledge of sources
impacting the study area; (ii) the comparison with the results reported by Vu et al. (2016), who
performed a PMF analysis of SMPS data collected in North Kensington (London urban
background); (iii) the shape of resulting profiles for both the particle number and volume
concentrations; (iv) the analysis of diurnal patterns; (v) the directional analysis using the polar plot
and CBPF; (vi) the correlations between the source contributions and the other air pollutants
monitored at the site or with weather variables, and (vii) the analysis of possible remote source
areas by applying the CWT model.

Six-factor solutions were extracted for both the seasons. The resulting factor profiles are presented
in Figures 6 and 7 for the warm and cold season, respectively. The factor profiles are expressed as:
(i) particle number concentrations and their DISP ranges; (ii) particle volume concentrations, and
(iii) explained variations showing how much of the variance (from 0 to 1) in the original dataset is
accounted for by each extracted factor. The figures also show the diurnal patterns and the polar
plots computed on the hourly-averaged contributions. Table 1 summarises the PMF results and
spectral characteristics, while Table 2 shows the Pearson correlation matrices with weather and air
quality variables. Selected PMF solutions were very stable: no errors or unmapped factors and few
swaps (none in summer and <7% in winter) were found in BS; no swaps or errors even at $dQ_{max}$=25
were found for DISP, i.e. solutions were affected by small rotational ambiguity and, therefore, their
interpretation can be considered robust.





DISP analysis is designed to explore the realistic bounds on the optimal (base run) PMF solutions
that do not result in appreciable increases in the $Q$ values (Brown et al., 2015). In this study, the
ranges calculated by DISP for the $dQ$=4 were used to assess the uncertainty boundaries associated
to the final PMF profiles, as suggested in Zikova et al. (2016) and Masiol et al. (2017). This
strategy is useful to better interpret the results, as the regions of spectra affected by high rotational
ambiguity are disclosed in the resulting profiles.

*3.3.1    Warm season*
*Factor 1* includes most of the particles in the nucleation range (<20 nm), exhibits a sharp mode in
the number distribution below the SMPS detection limit (14 nm) and makes the largest contribution
to the total PNC (31.6%, DISP range 31-36%) (Figure 6). However, its contribution to the volume
distribution is ~1%. Several studies report that particles in the nucleation range are emitted from the
aircraft engines (e.g., Anderson et al., 2005; Herndon et al., 2008; Kinsey et al., 2010; Mazaheri et
al., 2009;2013; Masiol and Harrison, 2014; Lobo et al., 2015) as well as from other anthropogenic
(e.g., Schneider et al., 2005; Chen et al., 2011; Cheung et al., 2012; Stevens et al., 2012; Kumar et
al., 2013;2014; Vu et al., 2015b) and natural (e.g., Kulmala et al., 1998; O'Dowd et al., 1998;1999;
Kulmala and Kerminen, 2008; Riccobono et al., 2014) sources. This factor does not show any
significant ($p < 0.05$) and strong ($r \geq |0.6|$) correlation with other measured species, but a weak ($|0.4|$
$\leq r < |0.6|$) correlation with Factor 2. Its diurnal variation (Figure 6) shows higher concentrations
between 6 am and 10 pm, and well agrees with the airport flight movements (Figure 2). The polar
plot analysis also indicates enhanced levels when winds > 2 m s$^{-1}$ blow from the airfield sectors
(SW). All these insights are consistent with the location of Heathrow, i.e. the most plausible
interpretation is related to the aircraft engine exhaust emissions. This interpretation is also
supported by Keuken et al. (2015), which shows that the PNSD in an area affected by emissions
from Schiphol airport (The Netherlands) is dominated by ultrafine (10-20 nm) particles. The large
contribution of this factor to the total PNC is not surprising if compared to the results reported for





the Los Angeles international airport by Hudda et al. (2014) (emissions from the airport increased
PNC 4- to 5-fold at 8–10 km downwind the airfield). Since the airport of Los Angeles and LHR
have comparable aircraft traffic, the quite high concentrations found in this study (on annual
average nucleation particles are ~10 times higher than those measured in North Kensington urban
background by Vu et al. (2016)) are consistent with the sampling location chosen in this study (~1.2
km to the airfield). In addition, this result also agrees with previous studies on the impacts of LHR
on local air quality; Carslaw et al. (2006) and Masiol and Harrison (2015) found comparable
percent contributions of LHR emissions on $NO_2$ levels in the study area (approx. 25-30%).
However, the lack of correlations with NO and $NO_2$ (tracers for aircraft emissions) is probably due
to the difference in the time resolution and the presence of several other sources of nitrogen oxides
in the area, such as the heavy traffic generated from the airport and from the nearby motorways.

*Factor 2* is made up of ultrafine particles in the nucleation-Aitken range (one main peak at 20-35
nm) and accounts for 28% (DISP 25-30%) of PNC; its contribution to the volume distribution is
low (~2%) and peaks at 22-45 nm and at 140-220 nm. Several insights seem to link this factor to
road traffic emissions: (i) the modal structure; (ii) the strong association with morning and evening
rush hours, and (iii) the significant increase for winds in the west and south-westerly sectors
consistent with emissions generated from local busy roads close to LHR, Tunnel Rd. and M25
motorway. A similar mode in the nucleation range has been extensively attributed to the size
distribution from road traffic (e.g., Vogt et al., 2003; Zhang et al., 2004; Ntziachristos et al., 2007;
Vu et al., 2015b) and the growth of nucleation particles from diesel vehicles (Mayer and Ristovski,
2007; Wehner et al., 2009). For example, Charron and Harrison (2003) reported that particles in the
range 30–60 nm show a stronger association with light-duty traffic at a traffic hotspot in central
London (Marylebone Rd.); Janhäll et al. (2004) reported an average particle size distribution
peaking at 15-30 nm during morning peak high traffic intensity in the city of Göteborg (Sweden),
which has a car fleet comparable to the UK; Ntziachristos et al. (2007) found a sharp mode at 20-30





nm in sampling from engine exhausts. In addition, PMF factors with similar modal structures were
found in other studies and were attributed to road traffic emissions: among others, Harrison et al.
(2011) linked a factor peaking at 20 nm to primary road traffic emissions near a major UK highway;
Masiol et al. (2016) measured PNSD in an international airport in Northern Italy during summer
and interpreted a factor with a clear mode at 35-40 nm as road traffic from the nearby city;
Beddows et al. (2015) and Vu et al. (2016) found traffic factors with modal diameter at around 30
nm in an urban background site in London (North Kensington); Sowlat et al. (2016) reported a
factor peaking at 20–40 nm in number concentration and at around 30–40 nm in volume
concentration in Los Angeles (US) and interpreted it as traffic tailpipe emissions. However, this
factor lacks significant positive correlations with primary road traffic tracers (nitrogen oxides,
eBC), while other studies have reported weak positive correlations with such species (Harrison et
al., 2011; Masiol et al., 2016; Vu et al., 2016; Sowlat et al., 2016). Similarly to factor 1, this latter
result may be due to the difference in the time resolution between chemical species and PNSD and
the presence of several sources of nitrogen oxides in the area.

*Factor 3* is mostly represented by 25–90 nm particles and contributes about 19% (17-21%) to the
total number concentration. It also shows a second mode below the SMPS detection limit (14 nm),
however, the DISP range clearly indicates that this part of the profile is affected by a large amount
of rotational ambiguity, so that the presence of this second mode should be interpreted with caution.
The volume concentration peaks at around 40–100 nm and 250–450 nm. The factor contribution is
higher during rush hours, but the morning peak occurs 1 h later than in factor 2. The wind
directionality shows increases for air masses blowing gently ($<4$ m s$^{-1}$) from W and for calm wind
periods, suggesting a quite local source; however, also an increase of concentrations is found for
higher wind regimes ($>6$ m s$^{-1}$) from the East (London). Factor 3 also shows significant positive
correlations with NO (0.43) and NO$_2$ (0.61). All these insights seem to point to an aged road traffic
source. This interpretation is also supported by Vu et al. (2016), who found a similar factor in





London (North Kensington) peaking at ~20–100 nm. In this context, several source apportionment
studies on PNSDs have attributed more than one factor to road traffic (e.g. Kasumba et al., 2009;
Thimmaiah et al., 2009; Harrison et al., 2011; Liu et al., 2014; Al-Dabbous and Kumar, 2015; Vu et
al., 2016; Sowlat et al., 2016). This result is not surprising in areas where heavy traffic is
widespread, as particles may undergo condensation, agglomeration, evaporation and dilution
processes and, consequently, they may change modal characteristics in time and space. Such
atmospheric processes are the main mechanisms reshaping PNSDs after primary exhausts are
emitted into the atmosphere and have been discussed in several studies (Shi et al., 1999; Kim et al.,
2004; Zhang et al., 2005; Zhou et al., 2005; Kulmala and Kerminen, 2008; Zhang et al., 2011;
Harrison et al., 2016).

*Factor 4* is made up of ultrafine particles over a wide range (50-200 nm with a clear mode at ~80
nm for PNC and 60-300 nm for PVC). The factor contributes 14% of PNC, but accounts for the
main percentage of the volume concentration (33%). This factor well correlates with gaseous
pollutants linked to combustion sources (mostly road traffic), i.e. NO (0.6), $NO_2$ (0.76), and non-
volatile primary pollutants, such as eBC (0.61), $NVPM_{2.5}$ (0.62) and EC (0.75). The factor also
strongly correlates with OC (0.84) and sulphate (0.75). The diurnal pattern shows two main peaks in
the morning and evening rush hours, but the concentrations recorded between the two maxima are
higher overnight than during daytime. This pattern suggests that both local emission sources and the
dynamics of the mixing layer may play a key role in shaping its diurnal cycle, i.e. emitted pollutants
undergo a wide dispersion within the expanded mixing layer during the daytime, while the drop of
the mixing layer occurring overnight restricts those pollutants to a layer close to ground level. The
polar plot indicates increased levels for wind calm or winds blowing from London (East sectors); in
addition, the factor is strongly negatively correlated with wind speed (-0.64).



All these insights suggest that Factor 4 represents the fingerprint of the London pollution. Several
studies carried out in London (Beddows et al., 2009;2015; Vu et al., 2016) and other megacities
(e.g., New York: Masiol et al., 2017) have reported similar results, all interpreting this source
profile as urban background (or urban accumulation mode). This source comprises both the solid
particle mode from traffic emissions (Harrison et al., 2011; Pant and Harrison, 2013; Dall'Osto et
al., 2012) and secondary species condensed upon pre-existing particles acting as condensation
nuclei, including secondary sulphate, nitrate and organic aerosols. Secondary sulphate is formed
through the atmospheric processing of local or distant $SO_2$ emissions (Kerminen et al., 2000) and
neutralisation with ammonia (Benson et al., 2011). Nitrate aerosol is formed through the oxidation
of $NO_2$ to nitrate and the consequent neutralization with ammonia (Seinfeld and Pandis, 2006) and
occurs during both daytime and night-time; however the semivolatile nature of ammonium nitrate,
makes its partitioning to the condensed-phase very weak. This behaviour also favours the
occurrence of negative artefacts in filter-based sampling, which may explain the lack of significant
correlations between the factor and the $PM_{2.5}$-bound nitrate (Table 2). On the contrary, the increase
of the intensity of factor 4 during the night-time and the significant association with $NO_2$ are highly
consistent with the chemistry driving the heterogeneous reactions of $N_2O_5$ and $NO_3$ on aerosol
surfaces (Bertram and Thornton, 2009; Brown and Stutz, 2012). In view of this, Dall'Osto et al.
(2009) reported that most nitrate particles in London are: (i) locally produced in urban locations
during nighttime; (ii) mainly present in particles smaller than 300 nm and (iii) internally mixed with
sulphate, ammonium, EC and OC.

Factors 5 and 6 make small contributions to PNC (4-7% and 1-4%, respectively), but are relevant
for the volume concentration (37% and 21%, respectively). Factor 5 shows a main accumulation
mode in number concentration at 110-250 nm and two more modes at ~30-70 nm and below 14 nm;
however, the latter two modes suffer of large rotational ambiguity and should be interpreted with
care. On the contrary, it exhibits a wide mode in volume concentration ranging from ~100 to ~500



nm. Factor 6 has two relevant modes in number concentration at 55-120 nm and 230-400 nm, and
two modes in volume concentration at 260-500 nm and 75-140 nm.

These factors still present two peaks corresponding to the rush hours, but the morning peak occurs
1-2 h earlier than in the road traffic-related factors, i.e. when ambient temperature reaches its daily
minimum. Both factors correlate well with secondary aerosol tracers (nitrate, sulphate, OC) and
non-volatile components (eBC, EC, NVPM$_{2.5}$), but Factor 6 exhibits much higher correlation
coefficients. Despite the polar plots indicating main wind directionality toward N-E sectors, the
analysis of air mass histories though the CWT model (Figure 8) clearly indicates likely continental
origin areas rather than local sources.

Vu et al. (2016) observed two factors in North Kensington with very similar modal structures, daily
patterns, correlations with PM$_{2.5}$-bound species and external source areas maps. Therefore, their
interpretation is confirmed also in this study, i.e. mixed secondary aerosol (Factor 5) and inorganic
secondary aerosol (Factor 6). Both factors are clearly originated from the continental Europe and
are consistent with a previous receptor modelling study carried out in a rural background site
representative of southern UK (Charron et al., 2013). Similar origin and formation mechanisms also
explain their strong correlation (0.75). Despite it is not reasonable extract more information from
these data due to the short period into account and the large uncertainty associated with back-
trajectory analysis, it can be observed that Factor 5 shows a wide source area all over the Central
Europe, while Factor 6 exhibits two distinct hotspots (Central and North-eastern Europe).

*3.3.2     Cold season*
The 6 factors identified during the cold period (Figure 7) are similar to those for the warm season.
*Factor 1* is composed of a high proportion of particles in the nucleation range with a sharp mode at
~15 nm. It accounts for 33% (32-35%) of PNC and less than 2% of PVC. The polar plot reveals





increased concentrations for moderate winds blowing from the airport sector and the diurnal pattern
is also compatible with the aircraft traffic. No statistically significant correlations are found with
any other monitored species. Therefore, Factor 1 may be attributed to the airport emissions related
to the aircraft engine exhausts emissions. As in the warm season, factor 1 is moderately correlated
with factor 2 (fresh road traffic, r=0.55), indicating a quite clear relationship between the two
sources.

*Factor 2* represents particles in the 15-35 nm range of number concentration, accounting for 35%
(33-37%) of total PNC. Its importance for volume concentration is modest (3%) with two modes at
30 and 200 nm. The diurnal pattern and the wind directionality are compatible with LHR as a
source and it shows a weak positive correlation with $NO_2$ (0.42) and a strong correlation with
nitrate (0.63). Despite its similarity and relationship with Factor 1 and the consequent similar
potential origin, Factor 2 may represent a different source: Factors 1 and 2 remain clearly separated
even at solutions down to 4 factors, demonstrating their structural robustness and the lack of
potential artefacts upon the PMF solution. Consequently, it can be concluded that they to not
represent over-resolved solutions (i.e. factor splitting). The most plausible interpretation for Factor
2 is therefore the same as for the warm season, i.e. fresh road traffic emissions. Furthermore, this
factor can be attributed to the road traffic generated by the airport and nearby major roads.

*Factor 3* includes most of the particles in the Aitken range and accounts for 19% (18-20%) of PNC.
It contribution to particle volume concentration is relevant (9%) with a main peak at around 100 nm
and a secondary peak at 400 nm. It presents two rush hours peaks and the polar plot reveals an
origin from the SW quadrant. However, as with the warm period, the wind directionality suggests
increases for slower wind regimes than the fresh road traffic factor and for more westerly sectors,
which are not compatible with the airfield location. Since factor 3 well correlates with a number of
other pollutants linked to primary emissions from road traffic (NO (0.51), $NO_2$ (0.81), $PM_{2.5}$ (0.53),





OC (0.79) and EC (0.83)), it represents a second road traffic factor, more affected by aging in the
atmosphere than factor 2.

Despite the wind regimes from NE sectors being poorly represented during the cold campaign,
*Factor 4* is the only one showing a possible origin from London and for calm wind periods. As with
the warm season, it is composed of a wide range of particles encompassing the Aitken and
accumulation modes (50 to 150 nm), while the peak in volume concentration is at 170 nm. The
diurnal pattern is clearly related to the mixing layer dynamics and the correlation analysis reveals
strong relationships with many species (NO, $NO_2$, eBC, $NVPM_{2.5}$, OC, EC, nitrate, ammonium and
potassium). Consequently, it is concluded that it represents the urban accumulation mode, whose
contribution to the total volume concentration is also similar to the warm season (33%). It is
interesting to note the large similarity with the urban accumulation mode found in the warm season,
from which it differs slightly only in the diurnal pattern (higher overnight) and in the presence of a
strong correlation with nitrate (r=0.88), due to the lesser extent of negative artefacts on $PM_{2.5}$ filter
samples.

The last two factors are interpreted as due to secondary aerosols. Their modal structures, their
contributions to total PNC and PVC, and their correlations with $PM_{2.5}$-bound species largely reflect
the results obtained for the warm period. However, the CWT maps (Figure 8) highlight different
source areas, i.e. the origin of the secondary aerosols is regional (UK and Northern Europe). In
addition, the presence of strong positive correlations with chloride may also indicate a contribution
from the transport of sea-salt aerosol.

**3.3    Comparison of *k*-means and PMF**
The cluster analysis revealed the presence of 5 characteristic PNSD shapes during both the seasons.
These spectra have been linked to potential sources in the study area, i.e. road traffic, airport



activities, biomass burning and secondary aerosol formation processes. However, the cluster
analysis is mostly driven by the size spectral regions with higher particle number concentrations,
i.e. it has the disadvantage of partitioning the single observations predominantly according to the
finest region of the size distribution. This limitation is well illustrated by the poor (almost null)
separation of clusters based on the particle volume distributions (all clusters showed quite similar
particle volume spectra). In addition, cluster analysis also has the disadvantage of linking each
cluster to a single source and does not easily account for PNSD resulting from the mix of two or
more different sources.

In contrast, the PMF analysis computed over the PNSD also accounts well for the sources with a
small impact on the number distribution, but having a larger influence on the particle volume size
distributions and, therefore, on the particle mass concentration. Despite the differences in the two
methods, some further information can be extracted by combining the results of cluster and PMF
analysis. Figure 9 shows the statistics of normalised PMF source contributions relating to each
single cluster. Generally, the two methods well agree for the "airport" source, pointing out how
much the airport-related emissions may shape the PNSD in the study area. For the warm period,
significantly higher (0.05 significance) PMF contributions of the airport factor (F1) are measured
for cluster 1, i.e. the airport fingerprint was well caught by both source apportionment methods.
During the cold season, the airport factor (F1) is high during both clusters 1 and 5. While cluster 5
presents significant high PMF contributions only for factor 1, cluster 1 also shows high
contributions of factor 2 (fresh road traffic). This result indicates that cluster 5 may be linked as the
typical PNSD spectra for airport emissions, while cluster 2 likely represents mixed emissions from
aircraft and airport-related traffic.

Results for fresh traffic emissions also agree between the two methods. Factors 2 exhibit the higher
normalised contributions to clusters 5 and 1 for the warm and cold period, respectively. However, in





winter it is evident that PNSDs grouped on cluster 1 are also strongly influenced by airport
emissions, probably due to the lower mixing layer height and, thus, a lesser dispersion in the
atmosphere.

Clusters 4 for both the periods show enrichments in the contributions for 4 PMF sources (aged road
traffic, urban accumulation and the two secondary aerosols). This further emphasises that cluster 4
represents the typical PNSD during daytime resulting from the mixing of different sources. In a
similar way, clusters 3 and 2 in the warm and cold periods, respectively, represent the typical
nighttime spectra, i.e. they exhibit similar partitioning over the PMF sources and similar daily
cycles.

**3.4     Analysis of a Large Regional Nucleation Event**
Regional photochemical nucleation episodes are regularly recorded in the Southern and Eastern
UK. Their general characteristics have been reported in a number of studies (e.g., Alam et al., 2003;
Charron et al., 2007;2008; Beddows et al., 2015; Vu et al., 2016) and can be summarised as
follows: (i) particle modality at around 20 nm; (ii) higher frequency around noon in association with
the peak in actinic flux intensities; (iii) clear seasonal cycles (higher average contribution levels in
the summer, from June to September); (iv) marked directionality from the westerly sectors,
reflecting maritime atmospheric circulation regimes, with high wind speed and low $PM_{2.5}$
concentrations.

A strong regional nucleation event occurred during the warm period sampling campaign (starting on
7th September at 1 pm UTC and lasting for about 12 h). Increases of PNC were almost
simultaneously recorded at Harlington and at Harwell, a national network rural background site
located approx. 60 km WNW of LHR and representative of the regional background levels of air
pollution across the Southern UK. The comparison of PNC time series at the two sites is provided





as Figure SI7. Figure 10 shows the contour plots of SMPS data recorded at Harlington between 7th
and 8th September as well as the hourly averaged concentrations of nucleation, Aitken and
accumulation particles, TEOM-FDMS $PM_{2.5}$ mass and the contributions of Factors 1 to 4 extracted
by the PMF. The figure also reports the hourly counts of number of clusters extracted by the *k*-
means analysis. The contour plot shows a typical "banana" shape with particle mode growing from
~20 nm (1 pm) to ~100 nm (overnight). The episode strongly influenced the PNSDs until around
midnight; however its effect is also visible over the first half of 8th September. The time series
(Figure 10) exhibits a clear peak in nucleation particles between 1 pm and 3 pm followed by peaks
of Aitken (3-11 pm) and accumulation mode (8 pm-2 am) particles. The back-trajectory analysis
(Figure 11) has revealed that the event occurred when north-westerly fresh (and clean) maritime air
masses were advected from the Atlantic. This is also supported by the $PM_{2.5}$ mass, which exhibited
a fast drop of concentrations just a few hours before the event, probably reducing the condensation
sink and facilitating nucleation.

Both atmospheric nucleation and aircraft engines are recognised to produce particles in the
nucleation range. The analysis of this single –but strong– episode gives insights into how much the
source apportionment results can potentially be affected by regional nucleation. This latter analysis
is possible because the wind directionality during the entire episode was from N sectors, i.e. the
contribution of LHR can be considered negligible.

The results of cluster analysis were just slightly affected by the event. Before the episode, the PNSD
spectra were mostly categorised as clusters 3 and 4 (urban background and daytime pollution,
respectively), while a few clusters (less than 1 h of observations) were categorised as "airport"
during the beginning of the episode. The growing of particles in the subsequent hours was then
identified as "fresh road traffic" (cluster 5) and "nighttime regional pollution" (cluster 2). In a





similar way, PMF results were slightly affected by the event, with a sharp increase of contribution
levels for factor 1 (airport) and, then, for factors 2 (fresh road traffic) and 3 (aged road traffic).

This episode was the main nucleation event recorded during the two sampling campaigns. Other
possible episodes also occurred (mostly during the warm season), but they were much less
significant and often hard to detect. This qualitative analysis points to some conclusions: (i)
regional photochemical nucleation events may have an effect on clustering and PMF results; (ii) the
effect may lead to an "additive" bias, mostly over the "airport" and "road traffic" factors and
clusters; (iii) the effect of regional nucleation events in the study area is largely overwhelmed by the
strength of local sources, but in other locations with more frequent nucleation events it may be more
important to identify and separate them.

**4        CONCLUSIONS**
The effect of airport emissions upon the particle number concentration and size distribution was
assessed at a site close to a major European airport (Heathrow) serving a megacity (London). The
conclusions to be drawn are:
•   Anomalously high particle number concentrations were recorded for the finest sizes (nucleation

849        <30 nm and Aitken nuclei 30-100 nm) if compared to an urban background site in London (N.

850        Kensington).

•   Polar plot analysis indicates that Heathrow is a strong potential source for $NO_2$, nucleation and

852        Aitken particles, but its contribution to the mass concentration of $PM_{2.5}$ and eBC is very small.

853        On the contrary, the London area seems to be a main source for PM and eBC.

•   The *k*-means cluster analysis has revealed that 20% of PNSDs are mostly shaped by airport

855        direct emissions, but particle size spectra are also strongly affected by other local sources

856        (mostly fresh and aged road traffic during daytime) and the reduction of mixing layer depth





(during nighttime). Typical PNSD spectra have been identified for nighttime and daytime
pollution as well. Such spectra are likely the result of multiple source mixtures.
• PMF analysis revealed that the fingerprint of Heathrow has a peculiar modal structure peaking
at <20 nm. The direct airport emissions account for 30-35% of total particles in both the
seasons. Such results are in line with percent estimations for $NO_2$ reported in previous studies.
• Other major contributors to PNC are fresh (24-36%) and aged (16-21%) road traffic emissions.
Despite both applied source apportionment methods failing to fully disaggregate the emissions
from the local traffic (including motorway) and traffic generated by the airport, results suggest
that road traffic sources may contribute to the total PNC more than Heathrow (40-56%).
However, making a clear distinction between the influence of traffic generated by the airport
from other road traffic is not feasible from this analysis.
• The fingerprint of London has a wide mode between 50-150 nm. This urban accumulation
mode accounts for around 10% of PNC and is the result of the advection of air masses from the
city. It is more evident overnight due to the drop of the mixing layer top, the subsequent
increase in air pollutants at ground level and the generation of nighttime secondary nitrate
aerosols.
• Secondary sources accounted for less than 6% in number concentrations but for more than 50%
in volume concentration. Long-range transport has a key role in advecting polluted air masses
from mainland Europe.

**ACKNOWLEDGEMNTS**
The authors gratefully acknowledge: (i) the European Union for funding the Marie Curie Intra-
European Fellowship for career development to M. Masiol through the project entitled 'Chemical
and Physical Properties and Source Apportionment of Airport Emissions in the context of European
Air Quality Directives (Project CHEERS, call: FP7-PEOPLE-2012-IEF, project no. 328542); (ii)
Heathrow Airport Ltd and Ricardo-AEA for supplying aircraft movement data and for the valuable





exchange of information and discussion, in particular Katherine Rolfe, Elizabeth Hegarty
(Heathrow), Brian Stacey (Ricardo-AEA) and David Vowles; (iii) DEFRA Automatic Urban and
Rural Network, and London Air Quality Network for providing pollutant data; (iv) Met Office and
BADC for weather data; (v) the NOAA Air Resources Laboratory (ARL) for the provision of the
HYSPLIT transport and dispersion model used in this publication; and (vi) Dr. Stefania Squizzato
(Clarkson University, USA) for the valuable exchange of information.






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

Contributions of aircraft arrivals and departures to ultrafine particle counts near Los Angeles
International Airport, Sci. Tot. Environ., 444, 347-355, 2013.
Hsu, H. H., Adamkiewicz, G., Houseman, E. A., Spengler, J. D., Levy and J.I.: Using mobile
monitoring to characterize roadway and aircraft contributions to ultrafine particle concentrations
near a mid-sized airport, Atmos. Environ., 89, 688-695, 2014.
Hudda, N., Gould, T., Hartin, K., Larson, T. V. and Fruin, S. A.: Emissions from an international
airport increase particle number concentrations 4-fold at 10 km downwind, Environ. Sci. Technol.,
1075 48, 6628-6635, 2014.
Hudda, N., Simon, M. C., Zamore, W., Brugge, D. And Durant, J. L.: Aviation emissions impact
ambient ultrafine particle concentrations in the greater Boston area, Environ.Sci. Technol., 50,
1079 8514-8521, 2016.
Hussein, T., Molgaard, B., Hannuniemi, H., Martikainen, J., Jarvi, L., Wegner, T., Ripamonti, G.,
Weber, S., Vesala, T. and Hameri, K.: Fingerprints of the urban particle number size distribution in
Helsinki, Finland: local vs. regional characteristics, Boreal Env. Res., 19, 1-20, 2014.
Janhäll S., Jonsson Å. M., Molnár P., Svensson E. A. and Hallquist M.: Size resolved traffic
emission factors of submicrometer particles, Atmos. Environ., 38, 4331-4340, 2004.
Kasumba, J., Hopke, P. K., Chalupa, D. C. and Utell, M. J.: Comparison of sources of submicron
particle number concentrations measured at two sites in Rochester, NY, Sci. Total Environ., 407,
1090 5071-5084, 2009.
Kelly, F. J. and Fussell, J. C.: Size, source and chemical composition as determinants of toxicity
attributable to ambient particulate matter, Atmos. Environ., 60, 504-526, 2012.



Kerminen, V. M., Pirjola, L., Boy, M., Eskola, A., Teinilä, K., Laakso, L., Asmi, A., Hienola, J.,
Lauri, A., Vainio, V. And Lehtinen, K.: Interaction between SO2 and submicron atmospheric
particles, Atmos. Res., 54, 41-57, 2000.
Keuken, M. P., Moerman, M., Zandveld, P., Henzing, J. S. and Hoek, G.: Total and size-resolved
particle number and black carbon concentrations in urban areas near Schiphol airport (the
Netherlands), Atmos. Environ., 104 132-142, 2015.
Kim, E., Hopke, P. K., Larson, T. V. and Covert, D. S.: Analysis of ambient particle size
distributions using unmix and positive matrix factorization, Environ. Sci. Technol., 38, 202-209,
1104    2004.
Kinsey, J. S., Dong, Y., Williams, D. C. and Logan, R.: Physical characterization of the fine
particle emissions from commercial aircraft engines during the aircraft particle emissions
experiment (APEX) 1 to 3, Atmos. Environ., 44, 2147-2156, 2010.
Kley, D., Kleinmann, M., Sanderman, H. and Krupa, S.: Photochemical oxidants: state of the
science, Environ. Pollut., 100, 19-42, 1999.
Knibbs, L. D., Cole-Hunter, T. and Morawska, L.: A review of commuter exposure to ultrafine
particles and its health effects, Atmos. Environ., 45, 2611-2622, 2011.
Kulmala, M., Toivonen, A., Mäkelä, J. M. and Laaksonen, A.: Analysis of the growth of nucleation
mode particles observed in Boreal forest, Tellus B, 50, 449-462, 1998.
Kulmala, M. and Kerminen, V.-M.: On the formation and growth of atmospheric nanoparticles,
Atmos. Res., 90, 132–150, 2008.
Kumar, P., Morawska, L., Birmili, W., Paasonen, P., Hu, M., Kulmala, M., Harrison, R. M.,
Norford, L. and Britter, R.: Ultrafine particles in cities, Environ.Int., 66, 1-10, 2014.
Kumar, P., Pirjola, L., Ketzel, M. and Harrison, R M.: Nanoparticle emissions from 11 non-vehicle
exhaust sources–A review, Atmos.Environ., 67, 252-277, 2013.
Lanzinger, S., Schneider, A., Breitner, S., Stafoggia, M., Erzen, I., Dostal, M., Pastorkova, A.,
Bastian, S., Cyrys, J., Zscheppang, A. and Kolodnitska, T.: Associations between ultrafine and fine
particles and mortality in five central European cities—Results from the UFIREG study, Environ.
Int., 88, 44-52, 2016.
Lee, D. S., Fahey, D. W., Forster, P. M., Newton, P. J., Wit, R. C. N., Lim, L. L., Owen, B., Sausen
and R.: Aviation and global climate change in the 21st century, Atmos. Environ., 43, 3520-3537,
1135    2009.
Liu, X., Wang, W., Liu, H., Geng, C., Zhang, W., Wang, H. and Liu, Z.: Number size distribution
of particles emitted from two kinds of typical boilers in a coal-fired power plant in China, Eng.
Fuels, 24, 1677-1681, 2010.
Liu, Z. R., Hu, B., Liu, Q., Sun, Y. and Wang, Y. S.: Source apportionment of urban fine particle
number concentration during summertime in Beijing, Atmos. Environ., 96, 359-369, 2014.





Lobo, P., Hagen, D. E. and Whitefield, P. D.: Measurement and analysis of aircraft engine PM emissions downwind of an active runway at the Oakland International Airport, Atmos. Environ., 61, 114-123, 2012.

Lobo, P., Hagen, D. E., Whitefield, P. D. and Raper, D.: PM emissions measurements of in-service commercial aircraft engines during the Delta-Atlanta Hartsfield Study, Atmos. Environ., 104, 237-245, 2015.

Lupu, A. and Maenhaut, W.: Application and comparison of two statistical trajectory techniques for identification of source regions of atmospheric aerosol species, Atmos. Environ., 36, 5607-5618, 2002.

Masiol, M. and Harrison, R. M.: Aircraft engine exhaust emissions and other airport-related contributions to ambient air pollution: A review, Atmos. Environ., 95, 409-455, 2014.

Masiol, M. and Harrison, R.M.: Quantification of air quality impacts of London Heathrow Airport (UK) from 2005 to 2012, Atmos. Environ., 116, 308-319, 2015.

Masiol, M., Vu, V. T., Beddows, D. C. S. and Harrison, R.M.: Source apportionment of wide range particle size spectra and black carbon collected at the airport of Venice (Italy), Atmos. Environ., 139, 56-74, 2016.

Masiol M., Hopke P. K., Felton H. D., Frank B. P., Rattigan O. V., Wurth M. J. and LaDuke G. H.: Source apportionment of $PM_{2.5}$ chemically speciated mass and particle number concentrations in New York City, Atmos. Environ.,148, 215-229, 2017.

Mazaheri, M., Johnson, G. R. and Morawska, L.: Particle and gaseous emissions from commercial aircraft at each stage of the landing and takeoff cycle, Environ. Sci. Technol., 43, 441-446, 2009.

Mazaheri, M., Bostrom, T. E., Johnson, G. R. and Morawska, L.: Composition and morphology of particle emissions from in-use aircraft during takeoff and landing, Environ. Sci. Technol., 47, 5235-5242, 2013.

Meyer, N. K. and Ristovski, Z.: Ternary nucleation as a mechanism for the production of diesel nanoparticles: experimental analysis of the volatile and hygroscopic properties of diesel exhaust using the volatilization and humidification tandem differential mobility analyser, Environ. Sci. Technol., 41, 7309-7314, 2007.

Ntziachristos, L., Ning, Z. Geller, M. D. and Sioutas, C.: Particle concentration and characteristics near a major freeway with heavy-duty diesel traffic, Environ. Sci. Technol., 41, 2223-2230, 2007.

O'Dowd, C. D., Geever, M., Hill, M. K., Smith, M. H. and Jennings, S. G.: New particle formation: Nucleation rates and spatial scales in the clean marine coastal environment, Geophys. Res. Lett., 25, 1661-1664, 1998.

O'Dowd, C., McFiggans, G., Creasey, D. J., Pirjola, L., Hoell, C., Smith, M. H., Allan, B. J., Plane, J. M. C., Heard, D. E., Lee, J. D., Pilling, M. J. and Kulmala, M.: On the photochemical production of new particles in the coastal boundary layer. Geophys. Res. Lett., 26, 1707-1710, 1999.

Ogulei, D., Hopke, P. K., Chalupa, D. C. and Utell, M. J.: Modeling source contributions to submicron particle number concentrations measured in Rochester, New York, Aerosol Sci. Technol., 41, 179-201, 2007.





Ostro, B., Hu, J., Goldberg, D., Reynolds, P., Hertz, A., Bernstein, L. and Kleeman, M. J.: Associations of mortality with long-term exposures to fine and ultrafine particles, species and sources: Results from the California Teachers Study Cohort, Environ. Health Perspect., 123, 549-556, 2015.

Paatero, P.: Least squares formulation of robust non-negative factor analysis, Chemom. Intell. Lab., 37, 23-35, 1997.

Paatero,, P. and Tapper, U.: Positive matrix factorization: a non-negative factor model with optimal utilization of error estimates of data values, Environmetrics, 5, 111-126, 1994.

Paatero, P., Hopke, P. K., Song, X. H. and Ramadan, Z.: Understanding and controlling rotations in factor analytic models, Chemom. Intell. Lab. Syst.. 60, 253-264, 2002.

Paatero, P., Eberly, S., Brown, S. G. and Norris, G. A.: Methods for estimating uncertainty in factor analytic solutions., Atmos. Meas. Tech., 7, 781-797, 2014.

Pant, P. and Harrison, R. M.: Estimation of the contribution of road traffic emissions to particulate matter concentrations from field measurements: a review, Atmos. Environ., 77, 78-97, 2013.

Petzold, A., Ogren, J.A., Fiebig, M., Laj, P., Li, S.M., Baltensperger, U., Holzer-Popp, T., Kinne, S., Pappalardo, G., Sugimoto, N. and Wehrli, C.: Recommendations for reporting "black carbon" measurements. Atmos. Chem. Phys., 13, 8365-8379, 2013.

R Core Team: R: A language and environment for statistical computing. R Foundation for Statistical Computing, Vienna, Austria, 2015. URL http://www.R-project.org/.

Reff, A., Eberly, S. I. and Bhave, P. V.: Receptor modeling of ambient particulate matter data using positive matrix factorization: review of existing methods, JAWMA, 57, 146-154, 2007.

Riccobono, F., Schobesberger, S., Scott, C. E., Dommen, J., Ortega, I. K., Rondo, L., Almeida, J., Amorim, A., Bianchi, F., Breitenlechner, M. And David, A.: Oxidation products of biogenic emissions contribute to nucleation of atmospheric particles, Science, 344, 717-721, 2014.

Rolph, G. D.: Real-time Environmental Applications and Display sYstem (READY) Website, http://www.ready.noaa.gov, NOAA Air Resources Laboratory, College Park, MD, 2016.

Salimi, F., Ristovski, Z., Mazaheri, M., Laiman, R., Crilley, L. R., He, C., Clifford, S. and Morawska, L.: Assessment and application of clustering techniques to atmospheric particle number size distribution for the purpose of source apportionment, Atmos. Chem. Phys., 14, 11883-11892, 2014.

Salma, I., Füri, P., Németh, Z., Balásházy, I., Hofmann, W. and Farkas, Á.: Lung burden and deposition distribution of inhaled atmospheric urban ultrafine particles as the first step in their health risk assessment, Atmos. Environ., 104, 39-49, 2015.

Sandradewi, J., Prévôt, A. S., Szidat, S., Perron, N., Alfarra, M. R., Lanz, V. A., Weingartner, E. and Baltensperger, U.: Using aerosol light absorption measurements for the quantitative determination of wood burning and traffic emission contributions to particulate matter, Environ. Sci. Technol., 42, 3316-3323, 2008.





Schneider, J., Hock, N., Weimer, S., Borrmann, S., Kirchner, U., Vogt, R. and Scheer, V.:
Nucleation particles in diesel exhaust: Composition inferred from in situ mass spectrometric
analysis, Environ. Sci. Technol., 39, 6153-6161, 2005.
Seinfeld, J. H. and Pandis, S. N.: Atmospheric Chemistry and Physics - From Air Pollution to
Climate Change, second ed., John Wiley & Sons, New York, 2006.
Shi, J. P. and Harrison, R. M.: Investigation of ultrafine particle formation during diesel exhaust
dilution, Environ. Sci. Technol., 33, 3730-3736, 1999.
Shi, L., Zanobetti, A., Kloog, I., Coull, B. A., Koutrakis, P., Melly, S. J. and Schwartz, J. D.: Low-
concentration PM2. 5 and mortality: Estimating acute and chronic effects in a population-based
study, Environ. Health Perspect., 124, 46-52, 2015.
Shirmohammadi, F., Sowlat, M. H., Hasheminassab, S., Saffari, A., Ban-Weiss, G. and Sioutas, C.:
Emission rates of particle number, mass and black carbon by the Los Angeles International Airport
(LAX) and its impact on air quality in Los Angeles, Atmos. Environ.,151, 82-93, 2017.
Sowlat M.H., Hasheminassab S. and Sioutas C.: Source apportionment of ambient particle number
concentrations in central Los Angeles using positive matrix factorization (PMF), Atmos. Chem.
Phys., 16, 4849-4866, 2016.
Squizzato, S. and Masiol, M.: Application of meteorology-based methods to determine local and
external contributions to particulate matter pollution: A case study in Venice (Italy), Atmos.
Environ., 119, 69-81, 2015.
Stein, A. F., Draxler, R. R, Rolph, G. D., Stunder, B. J. B., Cohen, M. D. and Ngan, F.: NOAA's
HYSPLIT atmospheric transport and dispersion modeling system, Bull. Amer. Meteor. Soc., 96,
1275    2059-2077, 2015.
Stevens, R. G., Pierce, J. R., Brock, C. A., Reed, M. K., Crawford, J. H., Holloway, J. S., Ryerson,
T. B., Huey, L. G. and Nowak, J. B.: Nucleation and growth of sulfate aerosol in coal-fired power
plant plumes: sensitivity to background aerosol and meteorology, Atmos. Chem. Phys., 12, 189-
1280    206, 2012.
Stohl A.: Trajectory statistics—a new method to establish source–receptor relationships of air
pollutants and its application to the transport of particulate sulfate in Europe, Atmos. Environ., 30,
1284    579-587, 1996.
Stohl, A.: Computation, accuracy and applications of trajectories- review and bibliography, Atmos.
Environ., 32, 947-966, 1998.
Stafoggia, M., Cattani, G., Forastiere, F., di Bucchianico, A. D. M., Gaeta, A. And Ancona, C.:
Particle number concentrations near the Rome-Ciampino city airport, Atmos. Environ., 147, 264-
1291    273, 2016.
Strak, M. M., Janssen, N. A., Godri, K. J., Gosens, I., Mudway, I. S., Cassee, F. R., Lebret, E.,
Kelly, F. J., Harrison, R. M., Brunekreef, B. and Steenhof, M.: Respiratory health effects of
airborne particulate matter: the role of particle size, composition, and oxidative potential-the
RAPTES project, Environ. Health Perspect., 120, 1183-1189, 2012.





Thimmaiah, D., Hovorka, J. and Hopke, P. K.: Source apportionment of winter submicron Prague aerosols from combined particle number size distribution and gaseous composition data. Aerosol Air Qual.Res., 9, 209-236, 2009.

USEPA: EPA Positive Matrix Factorization (PMF) 5.0 - Fundamentals and user guide. EPA/600/R-14/108, 2014

Vogt, R., Scheer, V., Casati, R. and Benter, T.: Onroad measurement of particle emission in the exhaust plume of a diesel passenger car, Environ. Sci. Technol., 37, 4070-4076, 2003.

Vu, T. V., Delgado-Saborit, J. M. and Harrison, R. M.: A review of hygroscopic growth factors of submicron aerosols from different sources and its implication for calculation of lung deposition efficiency of ambient aerosols, Air Quality, Atmos. Health, 8, 429-440, 2015a.

Vu, T. V., Delgado-Saborit, J. M. and Harrison, R. M.: Review: Particle number size distributions from seven major sources and implications for source apportionment studies, Atmos. Environ., 122, 114-132, 2015b.

Vu, T. V., Beddows, D. C. S., Delgado-Saborit, J. M. and Harrison, R. M.: Source Apportionment of the Lung Dose of Ambient Submicrometre Particulate Matter, Aerosol Air Quality Res., doi: 10.4209/aaqr.2015.09.0553, 2016

Yin, J., Harrison, R. M., Chen, Q., Rutter, A. and Schauer, J. J.: Source apportionment of fine particles at urban background and rural sites in the UK atmosphere, Atmos. Environ., 44, 841-851, 2010.

Yue, W., Stolzel, M., Cyrys, J., Pitz, M., Heinrich, J., Kreyling, W. G., Wichmann, H.-E.,Peters, A., Wang, S. and Hopke, P.K.: Source apportionment of ambient fine particle size distribution using positive matrix factorization in Erfurt, Germany, Sci. Total Environ., 398, 133-144, 2008.

Wang, Y., Hopke, P. K., Rattigan, O. V., Xia, X., Chalupa, D. C., Utell, M. J.: Characterization of residential wood combustion particles using the two-wavelength aethalometer, Environ.Sci. Technol., 45, 7387-7393, 2011.

Webb, S., Whitefield, P. D., Miake-Lye, R. C., Timko, M. T. and Thrasher, T. G.: Research needs associated with particulate emissions at airports, ACRP Report 6, Transportation Research Board, Washington, D.C., 2008.

Wehner, B., Uhrner, U., Von Löwis, S., Zallinger, M. and Wiedensohler, A.: Aerosol number size distributions within the exhaust plume of a diesel and a gasoline passenger car under on-road conditions and determination of emission factors, Atmos. Environ., 43, 1235-1245, 2009.

Wegner, T., Hussein, T., Hämeri, K., Vesala, T., Kulmala, M. and Weber, S.: Properties of aerosol signature size distributions in the urban environment as derived by cluster analysis, Atmos. Environ., 61, 350-360, 2012.

Wormhoudt, J., Herndon, S. C., Yelvington, P. E., Lye-Miake, R. C. and Wey, C.: Nitrogen oxide (NO/NO2/HONO) emissions measurements in aircraft exhausts, J. Propul. Power, 23, 906-911, 2007.




Zhang, K. M., Wexler, A. S., Zhu, Y. F., Hinds, W. C. and Sioutas, C.: Evolution of particle
number distribution near roadways. Part II: the 'Road-to-Ambient' process, Atmos. Environ., 38,
1350 6655-6665, 2004.
Zhang, K. M., Wexler, A. S., Niemeier, D. A., Zhu, Y. F., Hinds, W. C. and Sioutas, C.: Evolution
of particle number distribution near roadways. Part III: Traffic analysis and on-road size resolved
particulate emission factors, Atmos. Environ., 39, 4155-4166, 2005.
Zhang, R., Khalizov, A., Wang, L., Hu, M. and Xu, W.: Nucleation and growth of nanoparticles in
the atmosphere, Chem. Rev., 112, 1957-2011, 2011.
Zhou, L., Hopke, P. K., Stanier, C. O., Pandis, S. N., Ondov, J. M. and Pancras, J. P.: Investigation
of the relationship between chemical composition and size distribution of airborne particles by
partial least squares and positive matrix factorization, J. Geophys. Res.-Atmos., 110, D07S18, 2005,
doi:10.1029/2004JD005050.
Zíková, N., Wang, Y., Yang, F., Li, X., Tian, M. and Hopke, P. K.: On the source contribution to
Beijing PM 2.5 concentrations, Atmos. Environ., 134, 84-95, 2016.





**TABLE LEGENDS:**

**Table 1.**     Summary of PMF results for both seasons.

**Table 2.**     Results of Pearson's correlation analysis among extracted factor contributions and other measured variables recorded at different time resolutions. Only correlations significant at $p<0.05$ are reported, strong correlations ($\rho>|0.6|$) are highlighted in bold.

**FIGURE LEGENDS:**

**Figure 1.**     Map of LHR and sampling site (left) and map of the Greater London area (upper right). Wind roses calculated over the two sampling periods are also provided (bottom right). The location of some main potential sources is also highlighted: T1, T2, T3, T4 and T5 are the Heathrow terminals; TR= Tunnel Rd.

**Figure 2.**     Boxplots (a) and diurnal patterns (b) of the most important measured variables (and derived variables) during the two sampling periods. All valid data are used for computing boxplot statistics: Boxplot lines= medians, crosses= arithmetic means, boxes= 25th-75th percentile ranges, whiskers= ±1.5*inter-quartile ranges. Diurnal variations report the average levels as a filled line and the associated 95th confidence interval calculated by bootstrapping the data (n= 200). Outliers (data >99.5th percentile) were removed for computing the diurnal patterns. Hours are given in UTC. LHR traffic movements (bottom right plot) are reported as arrivals (dotted lines) and departures (solid lines). The offset between the seasons is largely due to daylight saving time (BST = UTC + 1) in the summer data.

**Figure 3.**     Statistics of size distribution spectra for particle number (red) and volume (blue) concentrations categorised by sampling periods and time of the day (daytime= 7am-7pm and nighttime=7pm- 7am local time). For the particle number spectra, solid lines represent the median concentrations, while shaded areas report the 1st-3rd quartile intervals. For the particle volume spectra, only medians are reported (dotted lines).

**Figure 4.**     Results of cluster analysis for the warm season data. Average cluster PNSD spectra (left) are reported as solid red lines along with: (i) their 10th, 25th, 75th and 90th percentile spectrum as shaded areas; (ii) the volume size distributions (dotted blue line); (iii) the hourly counts and (iv) the wind roses associated with each cluster.

**Figure 5.**     Results of cluster analysis for the cold season data. Average cluster PNSD spectra (left) are reported as solid red lines along with: (i) their 10th, 25th, 75th and 90th percentile spectrum as shaded areas; (ii) the volume size distributions (dotted blue line); (iii) the hourly counts and (iv) the wind roses associated with each cluster.

**Figure 6.**     Results of PMF analysis for the warm season data. Factor profiles are reported on the left as: (i) number concentration in solid red lines; (ii) their DISP ranges in shaded red areas; (iii) volume concentrations in dotted blue lines; (iv) explained variation in dashed grey lines. The plots on the centre report the normalised daily patterns calculated on the hourly-averaged factor contributions along with their 95th confidence intervals (n=200 bootstrap). The plots on the right show the polar plot analysis (normalised average factor contributions). SA=secondary aerosol.




**Figure 7.**   Results of PMF analysis for the cold season data. Factor profiles are reported on the left as: (i) number concentration in solid red lines; (ii) their DISP ranges in shaded red areas; (iii) volume concentrations in dotted blue lines; (iv) explained variation in dashed grey lines. The plots on the centre report the normalised daily patterns calculated on the hourly-averaged factor contributions along with their 95th confidence intervals (n=200 bootstrap). The plots on the right show the polar plot analysis (normalised average factor contributions). SA=secondary aerosol.

**Figure 8.**   CWT maps of the secondary aerosol-related factors for both the seasons. Map scales refer to the average factor contributions to the total variable (PNC).

**Figure 9.**    Comparison of k-means and PMF for the warm (upper plots) and cold (bottom plots) seasons. Boxplot statistics: lines= medians, crosses= arithmetic means, boxes= 25th-75th percentile ranges, whiskers= ±1.5*inter-quartile ranges.

**Figure 10.**   Analysis of the regional nucleation episode occurring on September 7th. The selected period is from 7 September  midnight to 8 September 4 pm. The plots represent (from upper to the bottom): (a) contour plots of SMPS data; (b) Concentrations of some measured species (Nucl= particles in the nucleation range 14-30 nm; Ait= particles in the Aitken Nuclei range 30-100 nm; Acc= particles in the accumulation range >100 nm; mass of $PM_{2.5}$); (c) Source contributions from PMF for the Factors 1, 2, 3 and 4; (d) hourly counts of number of clusters. The arrows in the (b) and (c) plots show the wind direction (arrow direction) and speed (proportional to arrow length).

**Figure 11.**   Backward air mass trajectories during the nucleation event. Dots indicate 24 h step times



**Table 1**. Summary of PMF results for both seasons.

| Factor number and interpretation | Particle Number Concentration | | Particle Volume Concentration | |
|---|---|---|---|---|
| **Warm season** (Aug-Sep 2014) | No. modes[a] (peak ranges[b]) | Percent contribution (DISP range) | No. modes[a] (peak ranges[b]) | Percent contribution |
| **Factor 1: Airport** | 1 (<20 nm) | 31.6 (30.8–36.2) | 2 (60–160 nm; <25 nm) | 1.2 |
| **Factor 2: Fresh road traffic** | 1 (20–35 nm) | 27.9 (24.7–30.2) | 2 (22–45 nm; 140–220 nm) | 1.7 |
| **Factor 3: Aged road traffic** | 1 (30–60 nm) | 18.9 (16.6–21.1) | 2 (40–100 nm; 250–450 nm) | 5.6 |
| **Factor 4: Urban accumulation** | 1 (50–150 nm) | 14.4 (13.8–18) | 1 (80–250 nm) | 33.2 |
| **Factor 5: Mixed SA** | 1 (110–250 nm) | 5.2 (3.6–6.9) | 1 (160–350 nm) | 37.4 |
| **Factor 6: Inorganic SA** | 2 (55–120 nm; 230–400 nm) | 2.1 (1.1–3.5) | 2 (260–500 nm; 75–140 nm) | 20.8 |
| **Cold season** (Dec 2014-Jan 2015) | | | | |
| **Factor 1: Airport** | 1 (<20 nm) | 33.1 (31.7–34.8) | 2 (160–350 nm; 15–25 nm) | 1.7 |
| **Factor 2: Fresh road traffic** | 1 (18–35 nm) | 35.2 (33.4–36.9) | 2 (22–45 nm; 150–300 nm) | 3.1 |
| **Factor 3: Aged road traffic** | 1 (28–60 nm) | 18.9 (17.9–19.7) | 2 (40–150 nm; 330–450 nm) | 8.7 |
| **Factor 4: Urban accumulation** | 1 (55–170 nm) | 7.6 (7.3–8.3) | 1 (100–250 nm) | 32.5 |
| **Factor 5: Mixed SA** | 2 (130–280 nm, <17 nm) | 2.3 (2.1–3.3) | 1 (170–400 nm) | 30.8 |
| **Factor 6: Inorganic SA** | 3 (17–28 nm; 55–100 nm, 250–400 nm) | 2.9 (2.4–3.9) | 2 (280–550 nm; 90–140 nm) | 23.3 |

(a) Only modes above the DISP ranges are shown; (b) Range endpoints are taken at approx. half the mode height.




**Table 2**. Results of Pearson's correlation analysis among extracted factor contributions and other
measured variables recorded at different time resolutions. Only correlations significant at $p<0.05$
are reported, strong correlations ($\rho>|0.6|$) are highlighted in bold.

| | Warm period | | | | | |
| | **Factor 1** | **Factor 2** | **Factor 3** | **Factor 4** | **Factor 5** | **Factor 6** |
| **Variables** | Airport | Fresh road traffic | Aged road traffic | Urban accumulation | Mixed SA | Inorganic SA |
|---|---|---|---|---|---|---|
| *Weather parameters (1 h-resolution time)* | | | | | | |
| **Solar irr.** | 0.12 | -0.15 | -0.24 | -0.26 | -0.24 | -0.28 |
| **Air temp.** | 0.25 | -0.21 | -0.37 | -0.1 | 0.1 | |
| **RH** | | 0.1 | 0.32 | 0.22 | 0.26 | 0.33 |
| **Wind speed** | 0.38 | | -0.47 | **-0.64** | -0.45 | -0.49 |
| *5 min-resolution time* | | | | | | |
| **Factor 1** | – | | | | | |
| **Factor 2** | 0.46 | – | | | | |
| **Factor 3** | 0.03 | 0.28 | – | | | |
| **Factor 4** | -0.17 | -0.04 | 0.47 | – | | |
| **Factor 5** | -0.15 | -0.06 | 0.21 | 0.56 | – | |
| **Factor 6** | -0.17 | -0.14 | 0.15 | 0.56 | **0.75** | – |
| **eBC** | -0.1 | -0.03 | 0.32 | **0.61** | 0.54 | 0.55 |
| **Delta-C** | | | -0.06 | -0.09 | -0.12 | -0.13 |
| *1 h-resolution time* | | | | | | |
| **NO** | | | 0.43 | **0.6** | 0.32 | 0.33 |
| **NO$_2$** | | 0.18 | **0.61** | **0.76** | 0.52 | 0.52 |
| **NO$_x$** | | 0.11 | 0.58 | **0.77** | 0.48 | 0.48 |
| **O$_3$** | 0.14 | -0.19 | -0.57 | -0.54 | -0.37 | -0.43 |
| **PM$_{2.5}$** | -0.23 | -0.24 | 0.13 | **0.61** | **0.63** | **0.77** |
| **NVPM$_{2.5}$** | -0.22 | -0.22 | 0.17 | **0.62** | **0.61** | **0.75** |
| **VPM$_{2.5}$** | -0.17 | -0.24 | | 0.42 | 0.54 | **0.65** |
| *1 day-resolution time PM$_{2.5}$-bound species* | | | | | | |
| **OC** | | | | **0.84** | **0.74** | **0.83** |
| **EC** | -0.47 | -0.54 | | **0.75** | 0.51 | **0.67** |
| **TC** | -0.45 | -0.44 | | **0.85** | **0.69** | **0.82** |
| **Chloride** | | | | | | |
| **Nitrate** | | -0.45 | | | **0.83** | **0.85** |
| **Sulphate** | | -0.57 | | **0.75** | 0.5 | **0.67** |
| **Oxalate** | | -0.47 | | 0.59 | **0.89** | **0.93** |
| **Sodium** | | | | | | |
| **Ammonium** | -0.44 | -0.52 | | 0.57 | 0.54 | **0.71** |
| **Potassium** | | -0.47 | | 0.46 | 0.5 | **0.66** |
| **Magnesium** | 0.5 | | | -0.53 | | |
| **Calcium** | | | | | | |






**Table 2**. Continued.

| Variables | Cold period | | | | | |
|---|---|---|---|---|---|---|
| | **Factor 1** | **Factor 2** | **Factor 3** | **Factor 4** | **Factor 5** | **Factor 6** |
| | Airport | Fresh road traffic | Aged road traffic | Urban accumulation | Mixed SA | Inorganic SA |
| *Weather parameters (1 h-resolution time)* | | | | | | |
| **Solar irr.** | | | | -0.11 | | |
| **Air temp.** | 0.38 | | -0.43 | **-0.67** | -0.5 | -0.59 |
| **RH** | | | 0.23 | 0.38 | 0.46 | 0.46 |
| **Wind speed** | 0.3 | | -0.49 | **-0.67** | -0.54 | **-0.61** |
| *5 min-resolution time* | | | | | | |
| **Factor 1** | – | | | | | |
| **Factor 2** | 0.55 | – | | | | |
| **Factor 3** | 0.24 | 0.54 | – | | | |
| **Factor 4** | -0.11 | 0.08 | 0.53 | – | | |
| **Factor 5** | -0.05 | 0.15 | 0.38 | **0.65** | – | |
| **Factor 6** | -0.09 | 0.08 | 0.39 | **0.7** | **0.81** | – |
| **eBC** | | 0.19 | 0.54 | **0.75** | 0.57 | **0.61** |
| **Delta-C** | | | 0.1 | 0.21 | 0.22 | 0.19 |
| *1 h-resolution time* | | | | | | |
| **NO** | -0.14 | | 0.51 | **0.81** | **0.62** | **0.63** |
| **NO$_2$** | 0.13 | 0.42 | **0.81** | **0.82** | **0.61** | **0.66** |
| **NO$_x$** | | 0.17 | **0.63** | **0.85** | **0.64** | **0.68** |
| **O$_3$** | | -0.29 | **-0.71** | **-0.78** | **-0.65** | **-0.7** |
| **PM$_{2.5}$** | -0.1 | 0.16 | 0.53 | **0.82** | **0.88** | **0.88** |
| **NVPM$_{2.5}$** | -0.11 | 0.16 | 0.53 | **0.82** | **0.85** | **0.85** |
| **VPM$_{2.5}$** | | | 0.19 | 0.39 | 0.49 | 0.48 |
| *1 day-resolution time PM$_{2.5}$-bound species* | | | | | | |
| **OC** | | | **0.79** | **0.79** | **0.76** | **0.8** |
| **EC** | | | **0.83** | **0.8** | **0.64** | **0.66** |
| **TC** | | | **0.81** | **0.8** | **0.73** | **0.77** |
| **Chloride** | | | | 0.58 | **0.82** | **0.85** |
| **Nitrate** | | **0.63** | **0.73** | **0.88** | **0.93** | **0.9** |
| **Sulphate** | | | | | **0.92** | **0.88** |
| **Oxalate** | | | | | **0.87** | **0.81** |
| **Sodium** | | -0.58 | **-0.74** | **-0.64** | | |
| **Ammonium** | | | **0.63** | **0.78** | **0.99** | **0.97** |
| **Potassium** | | | | **0.71** | **0.98** | **0.97** |
| **Magnesium** | | | | | | |
| **Calcium** | | | | | | |







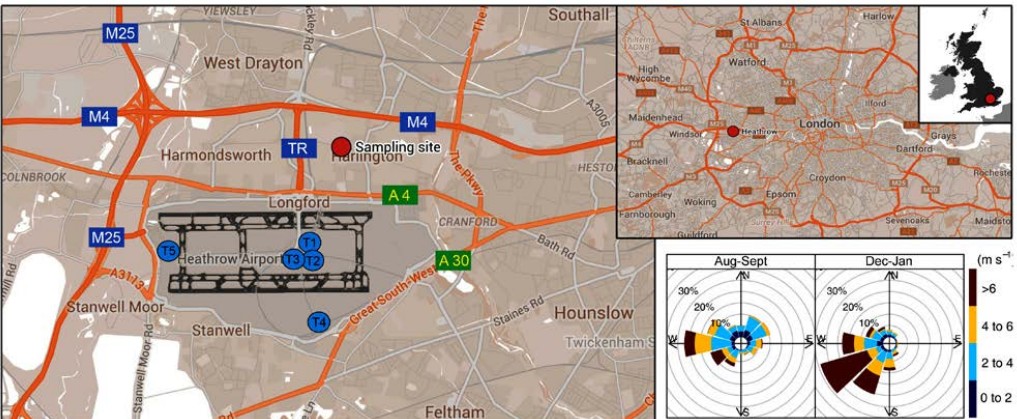


**Figure 1.** Map of LHR and sampling site (left) and map of the Greater London area (upper right). Wind roses calculated over the two sampling periods are also provided (bottom right). The location of some main potential sources is also highlighted: T1, T2, T3, T4 and T5 are the Heathrow terminals; TR= Tunnel Rd.

1465



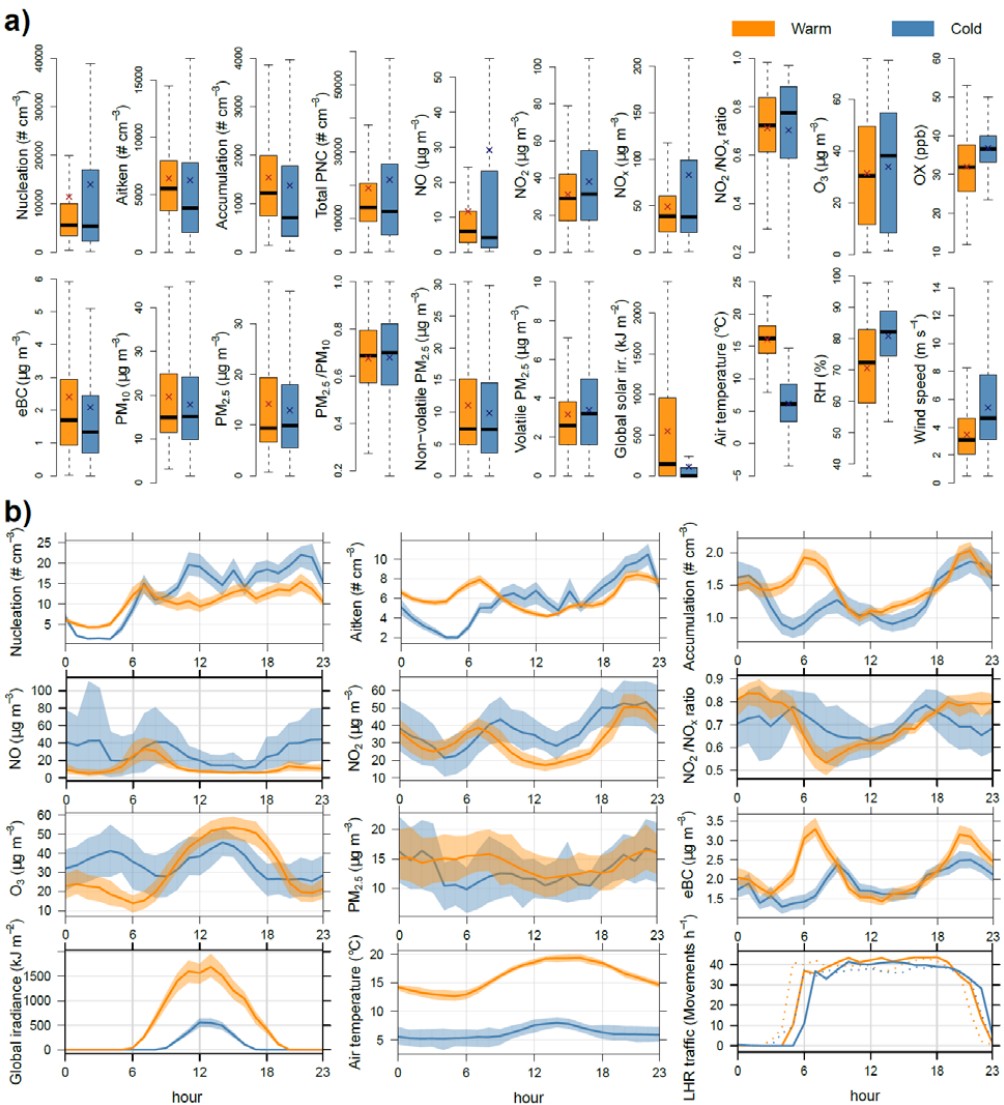

1466

**Figure 2.** Boxplots (a) and diurnal patterns (b) of the most important measured variables (and
derived variables) during the two sampling periods. All valid data are used for computing boxplot
statistics: Boxplot lines= medians, crosses= arithmetic means, boxes= 25th-75th percentile ranges,
whiskers= ±1.5*inter-quartile ranges. Diurnal variations report the average levels as a filled line and
the associated 95th confidence interval calculated by bootstrapping the data (n= 200). Outliers (data
>99.5th percentile) were removed for computing the diurnal patterns. Hours are given in UTC. LHR
traffic movements (bottom right plot) are reported as arrivals (dotted lines) and departures (solid
lines). The offset between the seasons is largely due to daylight saving time (BST = UTC + 1) in
the summer data.

1476





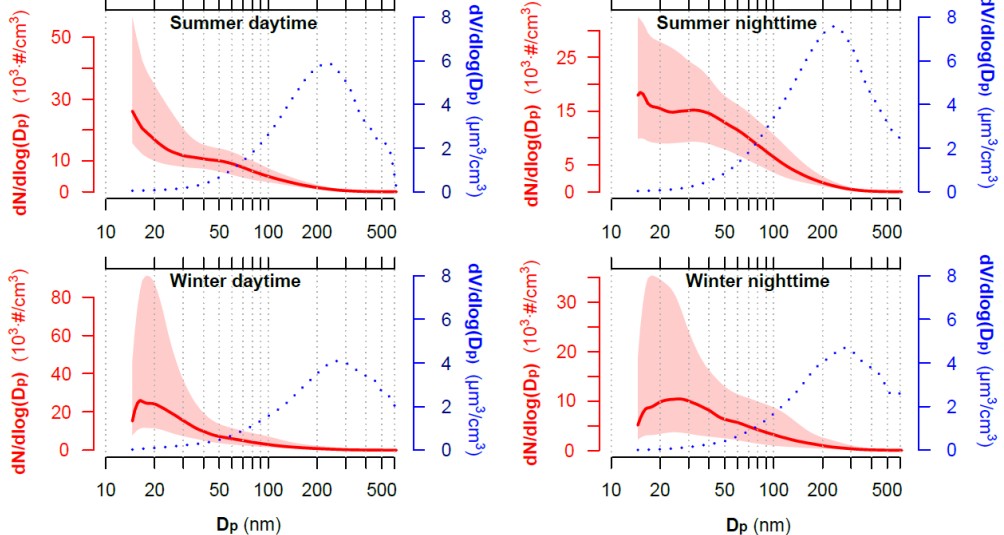

1477

**Figure 3.** Statistics of size distribution spectra for particle number (red) and volume (blue)
concentrations categorised by sampling periods and time of the day (daytime= 7am-7pm and
nighttime=7pm- 7am local time). For the particle number spectra, solid lines represent the median
concentrations, while shaded areas report the 1st-3rd quartile intervals. For the particle volume
spectra, only medians are reported (dotted lines).





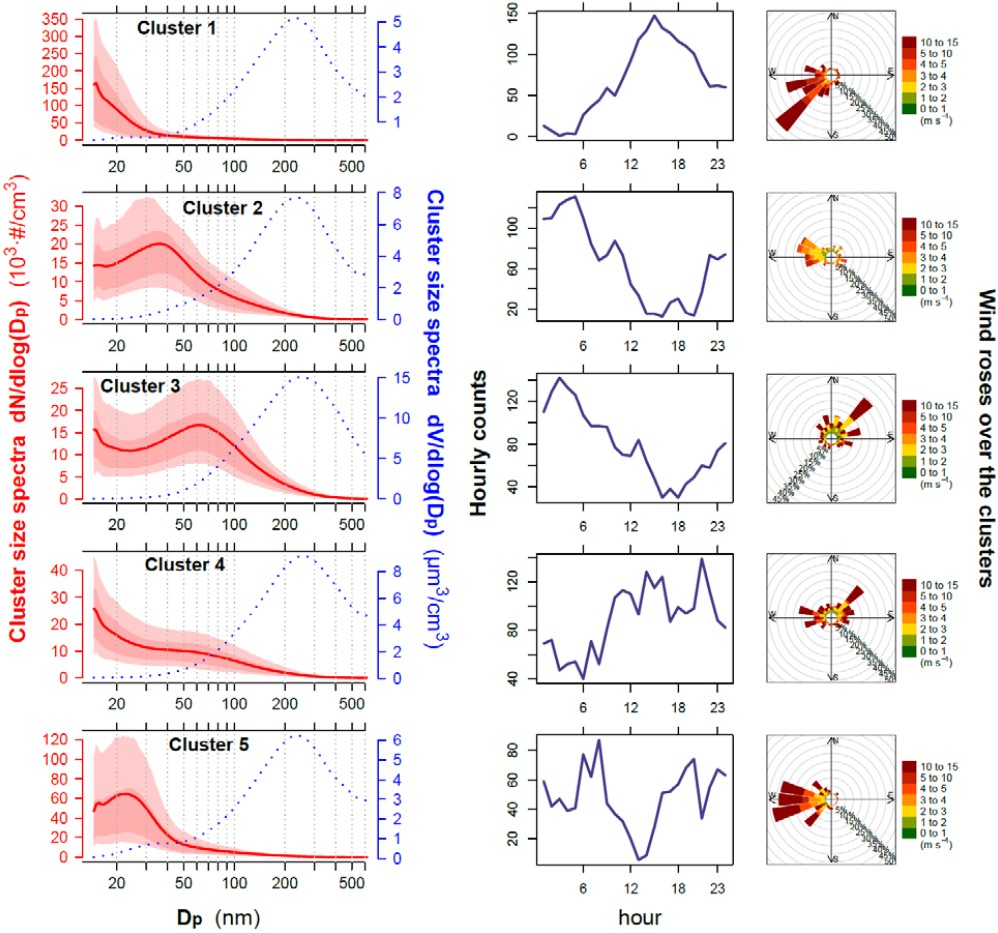

1483

**Figure 4.** Results of cluster analysis for the warm season data. Average cluster PNSD spectra (left)
are reported as solid red lines along with: (i) their 10th, 25th, 75th and 90th percentile spectrum as
shaded areas; (ii) the volume size distributions (dotted blue line); (iii) the hourly counts and (iv) the
wind roses associated with each cluster.



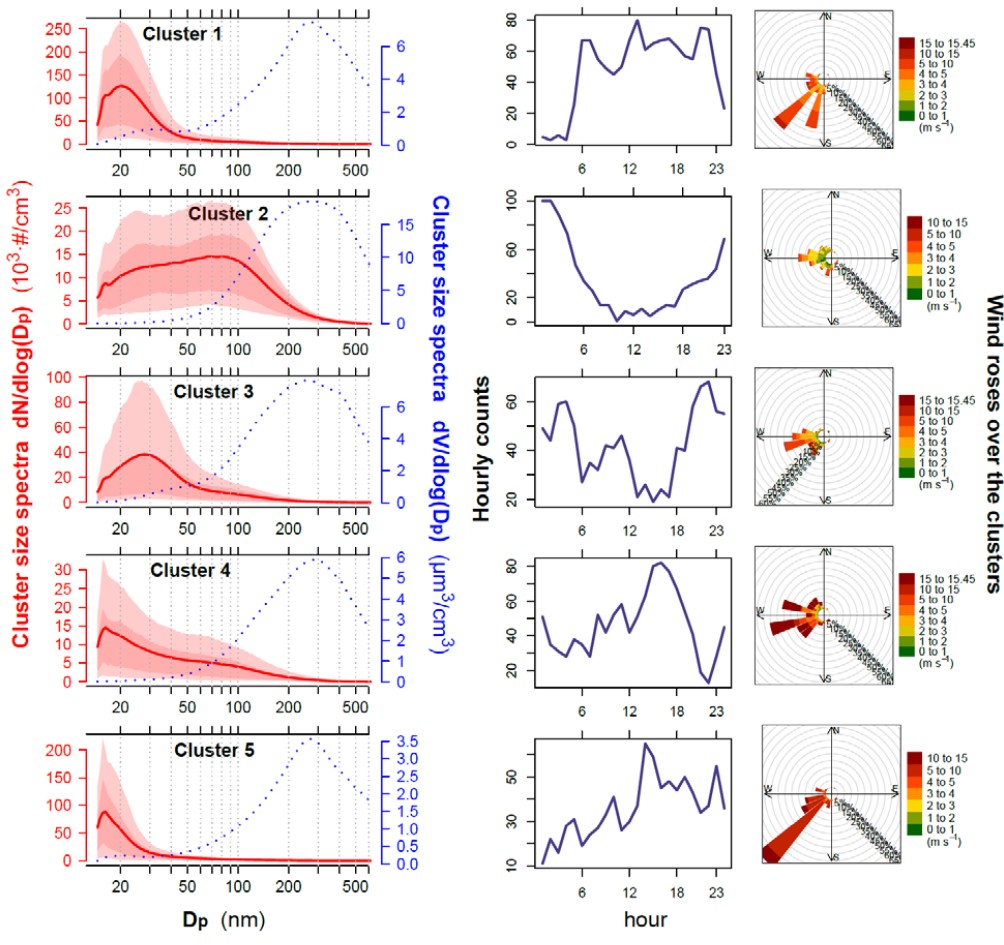


**Figure 5.** Results of cluster analysis for the cold season data. Average cluster PNSD spectra (left)
are reported as solid red lines along with: (i) their 10th, 25th, 75th and 90th percentile spectrum as
shaded areas; (ii) the volume size distributions (dotted blue line); (iii) the hourly counts and (iv) the
wind roses associated with each cluster.

















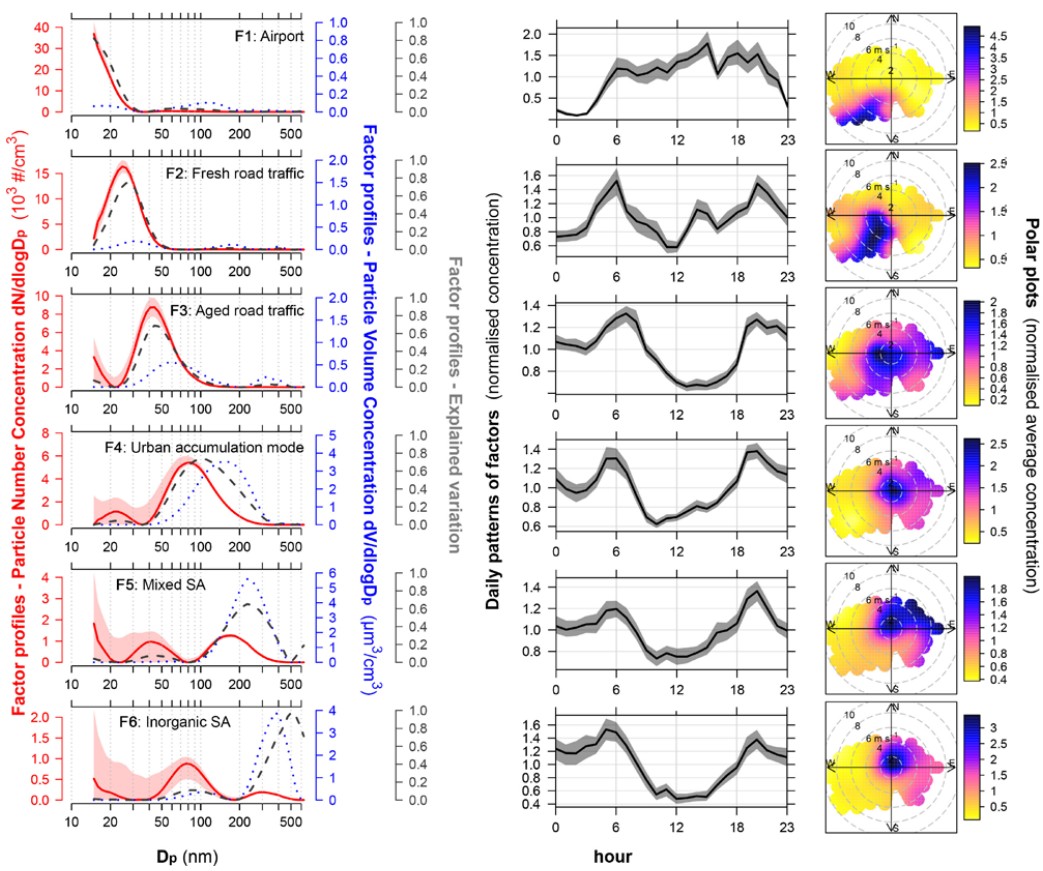

**Figure 6.** Results of PMF analysis for the warm season data. Factor profiles are reported on the left as: (i) number concentration in solid red lines; (ii) their DISP ranges in shaded red areas; (iii) volume concentrations in dotted blue lines; (iv) explained variation in dashed grey lines. The plots on the centre report the normalised daily patterns calculated on the hourly-averaged factor contributions along with their 95th confidence intervals (n=200 bootstrap). The plots on the right show the polar plot analysis (normalised average factor contributions). SA=secondary aerosol.





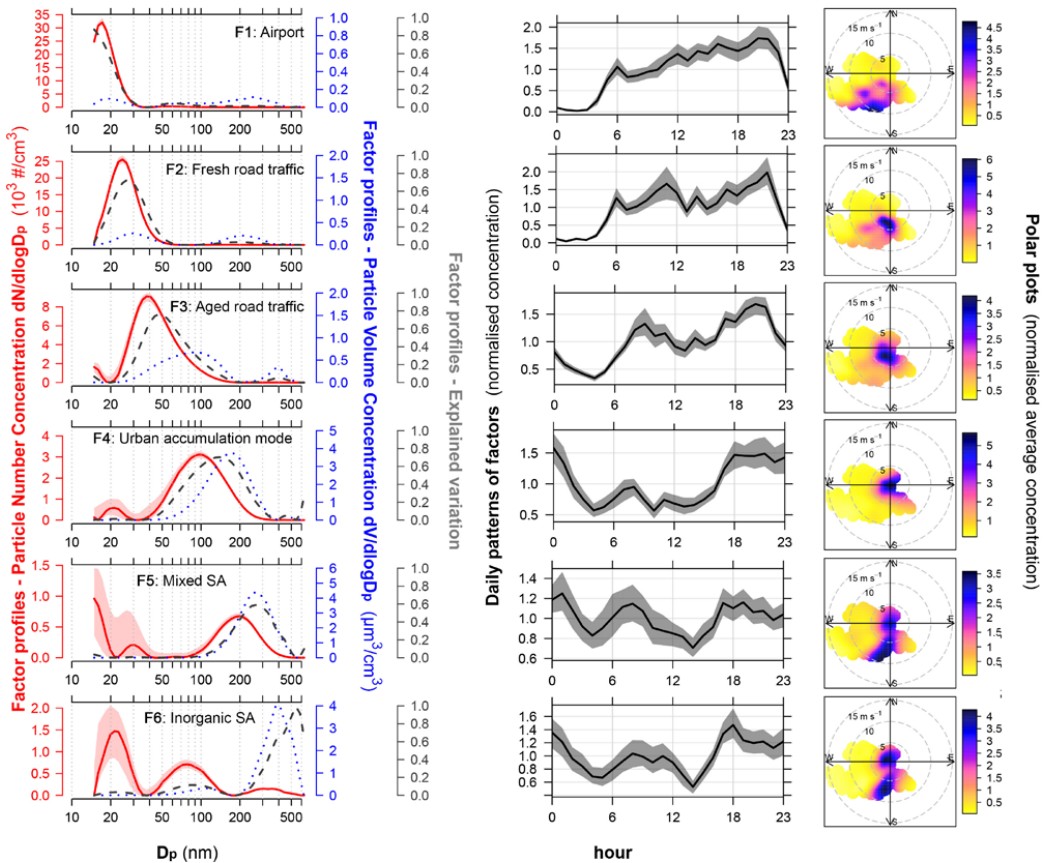

1511

**Figure 7.** Results of PMF analysis for the cold season data. Factor profiles are reported on the left
as: (i) number concentration in solid red lines; (ii) their DISP ranges in shaded red areas; (iii)
volume concentrations in dotted blue lines; (iv) explained variation in dashed grey lines. The plots
on the centre report the normalised daily patterns calculated on the hourly-averaged factor
contributions along with their 95th confidence intervals (n=200 bootstrap). The plots on the right
show the polar plot analysis (normalised average factor contributions). SA=secondary aerosol.



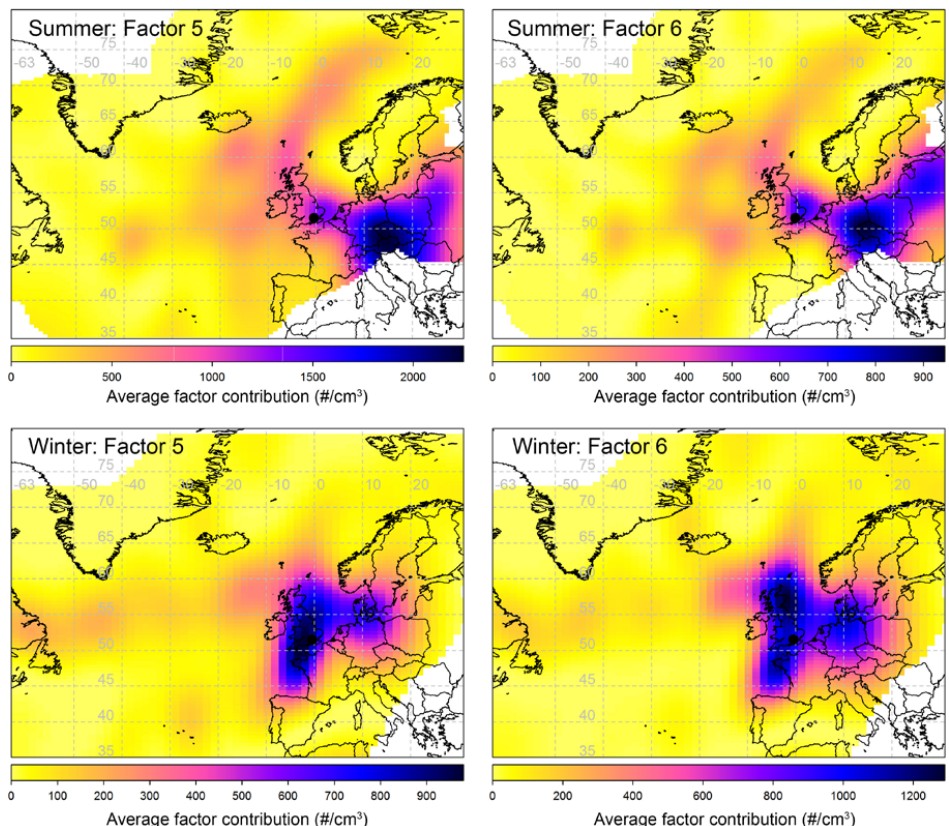


**Figure 8.** CWT maps of the secondary aerosol-related factors for both the seasons. Map scales refer
to the average factor contributions to the total variable (PNC).







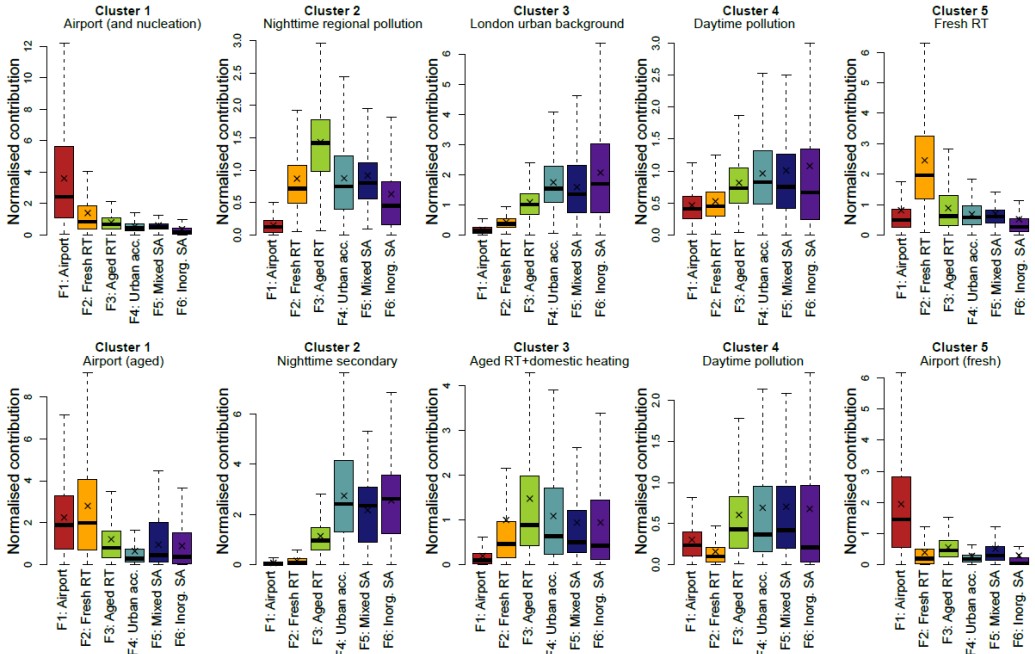


**Figure 9.** Comparison of k-means and PMF for the warm (upper plots) and cold (bottom plots)
seasons. Boxplot statistics: lines= medians, crosses= arithmetic means, boxes= 25th-75th percentile
ranges, whiskers= ±1.5*inter-quartile ranges.





**Figure 10.** Analysis of the regional nucleation episode occurring on September 7th. The selected period is from 7 September  midnight to 8 September 4 pm. The plots represent (from upper to the bottom): (a) contour plots of SMPS data; (b) Concentrations of some measured species (Nucl= particles in the nucleation range 14-30 nm; Ait= particles in the Aitken Nuclei range 30-100 nm; Acc= particles in the accumulation range >100 nm; mass of $PM_{2.5}$); (c) Source contributions from PMF for the Factors 1, 2, 3 and 4; (d) hourly counts of number of clusters. The arrows in the (b) and (c) plots show the wind direction (arrow direction) and speed (proportional to arrow length).



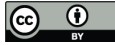

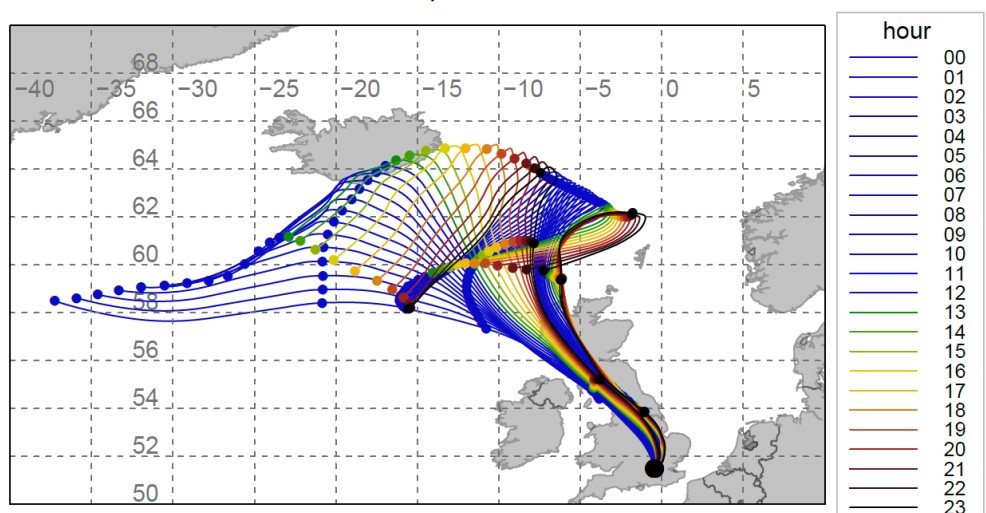


**Figure 11.** Backward air mass trajectories during the nucleation event. Dots indicate 24 h step times
