# Peer review of "Sources of Submicrometre Particles"

_Atmospheric Chemistry and Physics, 2017_

## Referee Comment (RC1) · Anonymous Referee #2 · 24 May 2017

The manuscript describes the results from a monitoring station close to LHR airport and uses two different analytical techniques to apportion or explain the particulate matter observed at the site. The data was collected during two intensives, nominally a warm and a cold season. The two analysis techniques are used to estimate the influence of the airport on the local particulate concentrations. It is clear the authors have analysed the data in detail and the use of the two techniques to interpret the data is good, as well as acknowledging the limitations.

Overall, I think the work is suitable for publication. I have 3 main suggestions/questions and some detailed comments below.

Main suggestions/questions:

1) I do not think the introduction is suitable as it does not provide a good background to

the manuscript (see details below). In addition, neither the introduction or the abstract mention the use of the regional nucleation event, which is really important for understanding a potential limitation in the two analysis techniques 2) I think a lot of the NOx and O3 details and graphs can be removed. The authors acknowledge that NOx has been studied in detail at the site before and the contributions from LHR to the site have been published. The paper itself is about the particulates, not air pollution in general. This would streamline the manuscript. 3) What version of AIM was used to analyse the data? If it was version 10, then no action is required. If it was version 9 however, then some work is needed. There is a significant difference between V9 and V10 in the diffusional correction algorithms. V9 over estimates the size of the correction needed and can lead to an over estimation of the nanoparticles, especially below 50nm. As the authors are dividing the particles into Aikten and Ultra fine modes (and the characteristics of these modes change with season), there is a need to check as it could lead to artefacts in the clustering.

Minor question:

Has the Aethelometer data been corrected according to Collaud Coen et al., 2010? I note the reported protocol as defined in Petzold et al., 2015. If not, it needs correcting especially if the authors want to draw BB aerosol information from it.

Details:

I question the relevance of some of the introduction to the manuscript. Lines 75 to 83 detail the pros and cons of a third runway, which I do not think adds to the manuscript as the study is about the impacts of Heathrow. It is fine to say the airport is planning to increase capacity, but as to why is not for this manuscript. In line 82, the use of the term 'Despite this' could be misinterpreted as questioning the UK government's decision.

Lines 85 – 104 Again, I think the context of this paragraph is not appropriate for the manuscript. The manuscript is about characterising the particulates close to Heathrow and not about questioning EU air quality standards as there is nothing in the manuscript

that takes this data and compares mortality or morbidity predicted from these measurements with those based on the EU standard method, for example. This section should be that despite the UK meeting the PM standard, it has been show UFP have health issues etc and because there is no network of measurements, campaigns such as this are key to assessing the sources and potential impacts of airports on the UFP. The last sentence should be removed from this section. As the authors point out (line 98), there is limited knowledge in this area, so what should be regulated and how? EU has standards on particulates from cars. New aircraft will soon be regulated on particulate number and mass. EU law requires restricting exposure to nano-particles in the work place. The UK has the clean air act. So there are policies in place, but more info in needed before guidelines and strategies can be applied.

Section 2.1 – study area and dates. With the exception of the first sentence, this section should be in the introduction. It is background and context material.

Lines 150-151, no need to repeat the dates here as they were just in the previous section.

Line 151-152, data from the site is QA etc. Which data? The authors or the data from the AURN site?

Lines 288 – 303, this reads like an introduction and is better suited in the introduction.

Section starting at line 309. This is not an acceptable way to present data and I don't see what ranking them in order shows. This also seems to repeat a lot of information already contained within figure 2a. Suggest removing most of that section and simply referring to fig 2a, M&H 2015 and that the data is representative.

Line 318, Suggest changing 'Since' to 'Analysis of the data showed it was not. . ...' and provide a reference for the Kruskal-Wallis analysis.

Line 324 and figure 3. Firstly, if the PNSD are shifting towards coarser modes, how can the PVSD be almost constant? The author's choice of wording is contradictory. Secondly, in the summer it appears the mode during the day could be anywhere between 20 and 50nm and during the night it is around 35nm. Is the statement backed up by fitting a double log-normal to the data? Thirdly, why aren't the percentiles for the PVSD presented if they are shown for the PNSD? Finally, why are the medians presented and not the means, which is more common? Does the later analysis use the means or medians? If the former, figure 3 should show means.

Line 330, meteorology plays a role.

Line 345, constant to 11pm (same value pre noon in summer)

Line 349, the accumulation mode peaks at around 10pm, which is inconsistent with the statement that it follows traffic which peaks at 5-6pm, for the M4 at least.

Line 351, I don't think the use of the word intermediate is appropriate. It is either traffic or aircraft or mixed, perhaps. Furthermore, this is interesting. Why is there a difference in winter and summer? If Version 9 of AIM was used, could this change your results if you try version 10?

Lines 355 – 381. I do not think this section needs to be in the manuscript. The title of the manuscript is about the sources of the particles near an airport. The NO2 is a) not showing any directionality and b) has been extensively studies already. The same is true for figures 2 a and b, the NOx data can be removed from there to streamline the paper.

Line 424 – I think cluster 2 can be bi modal as well, it certainly shows in the percentiles.

Line 429 – suggest changing to 'the POTENTIAL role. . ...'

Line 445 - again, is it mono-modal or bi-modal? The small bump at 14nm is in the median and percentiles.

Line 457 & figure 4. The hourly count for cluster 4 is very noisy. Where is the morning peak? The one just after 6am? I also disagree it is the mirror image of clusters 2 & 3

counts. It shows a broadly opposite trend. Suggest rewording.

Cluster 1 & 5 in the cold season – It seems a little odd that the count profile of cluster 1 matches more the profile of LHR than cluster 5, yet the suggestion is that cluster 1 is an aged LHR aerosol, and cluster 5 is fresh because short transit times (high wind speed). Surely a fresh emission will match more the LHR profile, while the process of aging will remove or diminish the effects of source? I don't think the author's conjecture is correct for cluster 5 being fresh emissions.

Cluster 3 cold season – can you back up your BB conjecture with the Aeth delta C data averaged over the same periods? If not, you should remove it.

Line 509 replace hump with mode.

Line 581 – The contribution to the NO2 levels at LHR are quoted here as 25-30%, but in the previous section (line 367) is was found to be 15-17% from another study. Was this study different to previous ones?

Line 637 – 50 – 200nm is not consistent with the authors definitions of UFP (<100nm)

Line 694 – 'to' missing from second sentence.

Line 711 – 3% is not modest, it is minimal.

Line 848 – suggest removing Anomalously . Being downwind of an airport is expected to lead to higher loadings.

---

## Referee Comment (RC2) · Anonymous Referee #1 · 25 May 2017

General comments:

This paper presents a study of different components of air pollution at London Heathrow Airport in two periods covering warm and cold environmental conditions. The authors report measurements of both particulate and gaseous pollutants and use k-means clustering and positive matrix factorisation in an attempt to apportion measured pollution to emissions sources and processes.

The dataset is extensive, the measurements are reliable and the analysis methods are appropriate. However, the presentation of the study is poor and therefore, the novel contribution of the paper is unclear. Below are major comments on the paper, after which follow specific line-by-line comments.

The paper has a number of weaknesses, and my opinion is that major revisions are

required before it is accepted for publication.

1. The literature review is incomplete and must be significantly improved. There have been numerous studies investigating the increased concentrations of UFPs close to airports, which must be included in the introduction. Some of these have been referenced in the discussion of results but inclusion of these references in Introduction is required to place this paper in the context of others and define the novel contribution.

2. The discussion of results is not well-structured, statements are not quantified sufficiently, and explicit references to figures are not included.

3. The section on the results from the k-means clustering (3.2) does not make any definitive conclusions, and it is limited in its contribution to novel science. Most of the results are justified by existing literature. Since that seems to be the case, the section is much longer than is appropriate for reporting routine results. Discussion should be limited to novel results, other results could be discussed in the SI.

4. The discussion of results from the k-mean clustering and PMF analysis is repetitive and many of the same references are used to infer the sources of particular clusters and factors. I would suggest that the discussion of these clusters and factors is combined in order to draw out stronger conclusions from the results (given the discussion in Section 3.3 noting good agreement between the two methods in identifying particular source signatures). The paper would be significantly improved by removing repetition within and the length of the Results section.

5. The paper is overly long and the results are not presented in a concise or coherent manner. There are several instances of repetition.

Specific comments:

Line 57: The statement that aviation growth will continue for the next decade cites a study from 8 years ago. Please use a more up to date reference.

Line 69: No reference for the 'indisputable' role of LHR in driving economic affluence

and vitality is given.

Line 75: Reference(s) for arguments in support of LHR expansion?

Line 82: Reference for Government approval of 3rd runway?

Line 106: Suggest present tense.

Line 108-117: Suggest description of methodology is moved to Section 2.

Line 119-121: Move to Acknowledgements.

Line 150: Repetition.

Line 165: Not all traffic is generated by the airport. Can the proportion be quantified?

Line 196: Suggest delete 'Classical', rephrase.

Line 223: Clarify the reason for deletion of data greater than 99.5th percentile. Is this for all measured quantities?

Line 261: Figures 1 and 2 are not adequately described in the text. If the data is not worth mentioning in the text, the figures should be moved to the SI.

Line 288: This paragraph would be more appropriate in the Introduction.

Line 312: Clarify meaning of 'intensive sampling'.

Line 330: Rephrase sentence 'Airport traffic undergoes...'

Line 340: The statement that nucleation particle concentrations 'drop to near zero overnight' is not substantiated by Figure 3. Statements discussing results must be quantified.

Section 3.1: References to figures should be made to aid interpretation.

Line 344: From Figure 2 it is not clear that accumulation mode concentrations have a peak corresponding to the morning rush hour.

Line 421-425: Repetition, rephrase.

Line 471: Clarify which clusters are being compared, 'their' is not sufficient. The comparisons between cluster 1 and cluster 5 are confused by the reference to the clusters from the warm period.

Line 482: What is the basis for this interpretation? This seems to be pure speculation.

Line 498: rephrase

Line 514: What is Q?

Line 513-530: These methodological details should be in the Methods section.

Line 765: Rephrase and quantify 'well agree'.

Line 788-796: Literature review should be in the Introduction.

Line 813: Quantify 'fast drop'.

Line 822: Quantify 'just slightly affected'.

Line 827: Quantify 'slightly affected'.

Line 843: 'Anomalously' implies that the measurements are flawed in some way. I do not believe that this is the case. Rephrase.

Line 863: Clarify 'the fingerprint of London'.

Figure 2: Check units of particle number concentrations – values are lower than typical ambient concentrations of 104 part/cm3 and are inconsistent with other figures.

Figure 10: This could be moved to the SI as discussion of it is very brief. Supplementary material: This is not referenced in the main text. It is also lacking any descriptive text and is just a collection of figures. This does not provide the reader with accessible or helpful information.

[Figure]

---

## Author Comment (AC1) · 26 Jul 2017

Journal: ACP MS No.: acp-2017-150 Title: Sources of Submicrometre Particles Near a Major International Airport Author(s): Mauro Masiol et al.

**RESPONSE TO REVIEWERS**

REFEREE #1 General comments: This paper presents a study of different components of air pollution at London Heathrow Airport in two periods covering warm and cold environmental conditions. The authors report measurements of both particulate and gaseous pollutants and use k-means clustering and positive matrix factorisation in an attempt to apportion measured pollution to emissions sources and processes.

The dataset is extensive, the measurements are reliable and the analysis methods are

appropriate. However, the presentation of the study is poor and therefore, the novel contribution of the paper is unclear. Below are major comments on the paper, after which follow specific line-by-line comments.

The paper has a number of weaknesses, and my opinion is that major revisions are required before it is accepted for publication.

Reply: We would like to thank referee #1 for his/her useful comments. All the points have been considered in this revised version of our manuscript. In particular, the paper was shortened and a number of references were added to the "Introduction" section or moved from other sections (according to the specific referee's comments). The novel contribution of this study with respect to the available literature is now explained and discussed in the "Introduction" section.

We have organised our replies by highlighting the referee comments in bold-italic font; our reply is in normal font.

1. The literature review is incomplete and must be significantly improved. There have been numerous studies investigating the increased concentrations of UFPs close to airports, which must be included in the introduction. Some of these have been referenced in the discussion of results but inclusion of these references in Introduction is required to place this paper in the context of others and define the novel contribution.

Reply: The "Introduction" section has been substantially amended. We have moved a whole paragraph including a number of up-to-date references from "Results" to "Introduction". We have also added several other references within the manuscript. We are aware that many other related studies are available in the literature. However, we have cited those that might be more relevant to the goals of our study. Our aim is that the current version of the "introduction" provides an adequate degree of information and allows the reader to understand the state-of-the-art on the topic.

In addition, some sentences were added to the "Introduction" to point out the novel con-
tribution of the paper with respect to the available literature. We are confident that our paper represents a novel contribution to the science: the literature offers many studies on the PNC and size distributions measured close or within airports. However, there are few papers using sophisticated chemometric tools to quantify and characterize the airport contribution to UFPs. Air quality close to large airports is potentially affected by many sources. The application of cluster and PMF analyses (and the comparison of their results) allows successful extraction of the profiles of individual sources. The comparison of the two methods (Section 3.3) allows the reader to understand the potentials and limits of each of these source apportionment techniques.

2. The discussion of results is not well-structured, statements are not quantified sufficiently, and explicit references to figures are not included.

Reply: The discussion of results has been extensively improved. Statements are now quantified. Explicit references to figures and/or tables are added throughout the manuscript. We have also shortened the sections related to the cluster analysis (see the referee's points 3 and 4 below).

3. The section on the results from the k-means clustering (3.2) does not make any definitive conclusions, and it is limited in its contribution to novel science. Most of the results are justified by existing literature. Since that seems to be the case, the section is much longer than is appropriate for reporting routine results. Discussion should be limited to novel results, other results could be discussed in the SI.

Reply: Cluster analysis extracts the most common modal spectra. It is not a novel science and it has been extensively applied to PNSD, particularly in London. However, we would like to stress that cluster analysis was never previously applied to 5 min resolved PNSDs in environments potentially affected by airport emissions. It represents the novel application of a well consolidated tool (clustering) to a specific case study (airports). In this way, we believe that its application is important for this study and generates new science useful for the scientific community. In addition, we believe that

**ACPD**
it is important to report the results of this technique in comparision with PMF analysis (as in manuscript Section 3.3). The PNSD spectra in airport-affected environments are very complex. Cluster analysis offers some simplification but was unable to resolve some of the sources.

We agree that the discussion of the cluster analysis results is long. However, we also believe that the cluster analysis gives important information and needs to be carefully discussed in the main text. Following the advice of the referee, the discussion of the cluster results was improved and shortened by about one third. In particular, we have merged the discussion of the warm and cold season results and have limited the main text to the main findings. We have also removed some misleading sentences and some statements not well supported by the data (as pointed out by the other referee). This makes the text shorter and has streamlined the discussion.

4. The discussion of results from the k-mean clustering and PMF analysis is repetitive and many of the same references are used to infer the sources of particular clusters and factors. I would suggest that the discussion of these clusters and factors is combined in order to draw out stronger conclusions from the results (given the discussion in Section 3.3 noting good agreement between the two methods in identifying particular source signatures). The paper would be significantly improved by removing repetition within and the length of the Results section.

Reply: As explained in the previous response to points 2 and 3, we have improved the sections on the cluster analysis. However, these two techniques are very different (as discussed in the text) and generate very different insights. Consequently, we believe that the sections cannot usefully be merged. On the contrary, we prefer to consider the outcomes of each method in turn and then consider what the combined data tells us about particle sources. We would like to point out that referee #2 does not criticise this part. Essentially, we found that in a location with so many particle source influences, clusters will never represent single source categories, whereas a well designed PMF is able to better extract the profiles of individual sources.

**ACPD**
5. The paper is overly long and the results are not presented in a concise or coherent manner. There are several instances of repetition.

Reply: Several sections of the manuscript have been carefully revised according to the comments of the two referees. The "Introduction" section was improved with additional references; the discussion of ancillary variables (particularily for NOx and O3) in the Section 3.1 "Overview of Data" was removed; two figures were moved to SI, while other figures were modified following advice from the referees; some statements in the "Results" were moved to the "Introduction"; the section on the results of the cluster analysis was shortened by  $\sim$ 30% by merging the results of the two seasons; statements were quantified (where possible) and references to figures and tables have been added throughout the text; we have removed several instances of repetition, errors/mistakes and misleading sentences.

Therefore, the manuscript underwent an overall shortenening and improvement, particularly in the discussion of the cluster analysis results. We believe that the manuscript is now well balanced and the discussion streamlined.

We recognise that the manuscript is long. However, we would like to stress that most of the literature offers studies based on the application of only one source apportionment method. This study has taken advantage of the synergy of the two methods (clustering and PMF) and their comparison. We believe that the careful discussion of outcomes from both the methods is useful and interesting for the scientific community and, therefore, needs to be preserved.

Specific comments: Line 57: The statement that aviation growth will continue for the next decade cites a study from 8 years ago. Please use a more up to date reference.

Reply: Up-to-date reference added. The estimation of the current trend was recently updated by the ICAO and is approx. 5.5%/y.

Line 69: No reference for the 'indisputable' role of LHR in driving economic affluence
and vitality is given.

Reply: Corrected. The whole paragraph was revised and shortened, according to the comments of the referee #2. In particular, the sentences related to the debate over the expansion of London Heathrow have been dropped off.

Line 75: Reference(s) for arguments in support of LHR expansion?

Reply: Corrected. The whole paragraph was revised and shortened, according to the comments of the referee #2. In particular, the sentences related to the debate over the expansion of London Heathrow have been dropped off.

Line 82: Reference for Government approval of 3rd runway?

Reply: A reference from the UK Department for Transport has been added.

Line 106: Suggest present tense. Reply: Done.

Line 108-117: Suggest description of methodology is moved to Section 2.

Reply: This paragraph has been modified and shortened. Here, we only present the goals of the study and briefly list the adopted source apportionment methods. In addition, we have added two paragraphs pointing out the novelty of the study compared to the current literature and the secondary goal related to the analysis of the effects of a regional nucleation event.

Line 119-121: Move to Acknowledgements.

Reply: Done. "Classical" substituted with "Routine". These are pollutants routinely measured at DEFRA air quality sites.

Line 150: Repetition.

Reply: Done. The whole Materials and Methods section has been revised. In particular, the original subsection 2.1 was modified and moved to the "Introduction". All the details about the sampling campaigns are now in the new subsection 2.1 "Experimen-

ACPD
tal".

Line 165: Not all traffic is generated by the airport. Can the proportion be quantified?

Reply: Sentence modified accordingly. Unfortunately, this is not quantifiable in this study. Our previous investigations (Masiol and Harrison, AE116, 2015) pointed out that the study of the differences (deltas) among multiple sampling sites across the study area is very useful to apportion the different sources, including traffic. However, just one site was used in this study.

The amended text: "The site is also affected by pollutants arising from the large volumes of road traffic in London, from the local road network as well as those generated by the airport. Tunnel Rd., the main access to LHR from the M4 motorway lies 800 m west, as well as the nearby M4 (640 m north) and M25 ( $\sim$ 3.5 km east) motorways, major roads (Bath Rd, part of A4, passes 900 m south; A30 lies 2.8 km SE). The village of Harlington ( $\sim$ 400 m west) and advection of air masses from the conurbation of London are other potential external sources."

Line 196: Suggest delete 'Classical', rephrase.

Reply: Done.

Line 223: Clarify the reason for deletion of data greater than 99.5th percentile. Is this for all measured quantities?

Reply: Only SMPS data have been handled in this way. Other data from the Harlington site are already checked and ratified by DEFRA. We have added some sentences explaining our choice. Essentially, we recognised some extreme events, which have been interpreted as outliers and/or instrumental errors. Most of them were linked to unusual activities or probable instrumental issues. Anyway, the modal structure of the PMF profiles with or without the removed data does not change. The main reasons driving this choice were: (i) a general improvement in the stability of PMF solutions; (ii) the decrease of uncertainty assessed by BS and DISP; and (iii) the decrease of scaled
residuals over +/-3, which, essentially, are records not well modeled by the PMF.

The amended text: "An in-depth analysis of the dataset revealed a few records with anomalously high PNC, which were likely related to probable instrumental issues, extreme weather conditions (e.g., high wind gusts, heavy rain striking the inlet), or infrequent local emissions, e.g., maintenance, painting and recreational activities (including fires) on the playground where the site is located, road maintenance close the site and probable short-term parking of high-emission vehicles near the site. Since this study aims to investigate the overall contributions of LHR, all data are used for descriptive statistics, but data greater than the 99.5th percentile were further removed for explorative, cluster and PMF analyses. This data exclusion successfully removed the extremely high events occurring during the sampling campaigns and significantly improved the stability and physical meaning of PMF solutions."

Line 261: Figures 1 and 2 are not adequately described in the text. If the data is not worth mentioning in the text, the figures should be moved to the SI.

Reply: Figure 1 (maps of area and sampling site) has been moved to SI (Figure SI1). Figure 2 (boxplots and diurnal patterns) was amended: now only some plots of part B (diurnal patterns) are shown in the main text, while part A (boxplots) was moved to SI (Figure SI3) as well as the plots of all the diurnal patterns (Figure SI4). We want to maintain some plots of the part B in the main text: we believe that showing the diurnal patterns of such important variables (PNC over the three size ranges, BC, solar irradiation and airport traffic) is really useful to the reader for helping the interpretation of results. These patterns can be, therefore, quickly compared with those reported for the clusters or the PMF factors.

Line 288: This paragraph would be more appropriate in the Introduction.

Reply: Done. This part was moved to the "Introduction" (also according to the main point 1 and the referee #2).

**ACPD**
Line 312: Clarify meaning of 'intensive sampling'.

Reply: Amended: "intensive" removed from the text.

Line 330: Rephrase sentence 'Airport traffic undergoes: : :'

Reply: Sentence modified accordingly: "During nighttime, airport traffic is restricted to limit noise and community disturbance: ....".

Line 340: The statement that nucleation particle concentrations 'drop to near zero overnight' is not substantiated by Figure 3. Statements discussing results must be quantified.

Reply: Amended. The whole paragraph was amended and the discussion of the patterns is now quantified.

Section 3.1: References to figures should be made to aid interpretation.

Reply: Amended. The whole text was carefully revised to link sentences and discussion to the appropriate figures and tables.

Line 344: From Figure 2 it is not clear that accumulation mode concentrations have a peak corresponding to the morning rush hour.

Reply: The morning peak is only evident during the warm season. The sentences have been amended consequently: "Accumulation particles also present the morning (6-8 am) and evening (6-11 pm) rush hour peaks during the warm season, but only the evening peak (from 6 pm to the night) was found in the cold season (Figure 2). Generally, the evening peaks start around 6 pm, which is consistent with the peak of traffic (Figure SI5) but they extend late in the evening and night probably because the drop of the mixing layer top and the consequent concentration of pollutants close to the ground level."

Line 421-425: Repetition, rephrase.
Reply: The discussion of the cluster analysis results was strongly modified and shortened (see reply to the referee's main points 2-5). We believe that there are no repetitions in its current form.

Line 471: Clarify which clusters are being compared, 'their' is not sufficient. The comparisons between cluster 1 and cluster 5 are confused by the reference to the clusters from the warm period.

Reply: Done.

Line 482: What is the basis for this interpretation? This seems to be pure speculation.

Reply: We have removed the sentences and we have improved the further discussion of the two clusters in Section 3.3 (comparison of cluster analysis and PMF). This was added to the text as: "The reasons driving the split of the spectra likely to be affected by LHR into two clusters during the cold season are unclear. A further comparison of the cluster and PMF results will help in interpreting this outcome."

Line 498: rephrase

Reply: Done. The whole section has been enhanced.

Line 514: What is Q?

Reply: It is the PMF objective function. In PMF, factor elements are constrained, therefore no sample can have a significantly negative contribution. PMF allows each data value to be individually weighted. This feature allows analysts to adjust the influence of each data point, depending on the confidence in the measurement. The PMF solution minimizes the object function Q via a conjugate gradient algorithm, based upon the estimated data uncertainties (or adjusted data uncertainties). A quick review of the PMF and how the objective function works is reported in Brown et al. (STOTEN 518-519, 2015) or in the PMF user's manual. Factor contributions and profiles are therefore derived by the PMF model minimizing the objective function (Q). **ACPD**
This paragraph was moved to the "Materials and Method" section (subsection 2.2), according to the following point of this referee. We have only included its name "objective function" into the text. We did not provide much detail on the science behind the PMF analysis, as it is a very well established technique. However, we noticed that most of the literature on PMF does not provide sufficient data to describe how the model was run and what diagnostics have been taken into account. In our opinion, this is a serious omission, as it is always important to know if the model is performed with the optimal criteria and if the model returns acceptable diagnostics. Consequently, we want to keep this (and other) brief technical information in the main text to provide all the information needed by the readers/PMF analysts who want to fully reproduce/check the method. This information is comparable to a QC/QA applied to the PMF analysis.

Line 513-530: These methodological details should be in the Methods section.

Reply: Done. Moved to "Materials and Methods" (subsection 2.2).

Line 765: Rephrase and quantify 'well agree'.

Reply: "well agree" was deleted. The normalised contributions are now quantified in the text and some statistical tests have been also applied in the discussion of results.

The new paragraph is: "For the warm period, significantly higher (0.05 significance) PMF contributions of the airport factor (F1) are measured for Cluster 1 (average normalised contribution  $\sim$ 3.5). This result indicates that the airport fingerprint was well captured by both source apportionment methods. During the cold season, the airport factor (F1) is significantly higher for both clusters 1 and 5 (average normalised contributions of  $\sim$ 2 and  $\sim$ 3, respectively). While Cluster 5 presents significant high PMF contributions only for factor 1, Cluster 1 also shows high contributions of factor 2 (fresh road traffic). This result indicates that Cluster 5 may be linked as the typical PNSD spectra for airport emissions, while Cluster 2 likely represents mixed emissions from aircraft and airport-related traffic. A close analysis of wind roses for the two clusters in the cold season (Figure 4) reveals that Cluster 5 occurs at significantly higher wind

**ACPD**
speed regimes than Cluster 1 (Mann-Whitney-Wilcoxon test at 0.05 significance level), i.e. average wind speeds of 8.3 and 5.9 m s-1, respectively. As a consequence, the different wind regimes may be likely responsible of the split between the two clusters.".

Line 788-796: Literature review should be in the Introduction.

Reply: Done. The whole paragraph was moved to the "Introduction".

Line 813: Quantify 'fast drop'.

Reply: Quantified. The drop was -30  $\mu$ g/m3 in 3 hours. Amended sentence: "This is also supported by the PM2.5 mass, which exhibited a rapid fall in concentrations just a few hours before the event (-30  $\mu$ g m-3 in 3 hours, i.e. from 40  $\mu$ g m-3 at 6 am to 10  $\mu$ g m-3 at 9 am, Figure 9), probably reducing the condensation sink and facilitating nucleation."

Line 822: Quantify 'just slightly affected'.

Reply: Quantified. The whole paragraph has been re-written and now presents a large number of quantified statements and explicit references to figures.

Amended paragraph: "The results of cluster analysis were affected by the event. Before the episode, the PNSD spectra were mostly categorised as Clusters 3 and 4 (urban background and daytime pollution, respectively), i.e. the clusters mostly recorded under north-easterly wind regimes (Figure 3). About 50% and 30% of the clusters were then categorised as "airport" in the first and second hour of the episode, respectively (Figure 9). Since the wind directionality is inconsistent with an origin from the airfield, this categorisation is likely the result of the nucleation event. The growth of particles in the hours after the beginning of the event has further driven the cluster results: (i) about 60-80% of PNSDs were categorised as "fresh road traffic" (Cluster 5) after 2-3 hours, and (ii) 80-100% of PNSDs were clustered as "nighttime regional pollution" (Cluster 2) after 4-6 hours. In a similar way, PMF results were affected by the event (Figure 9), with a sharp increase of contribution levels for: (i) factor 1 (airport) from 1.5 x 103 particles ACPD
cm-3 at noon to 13.3 x 103 particles cm-3 at 2 pm; (ii) factor 2 (fresh road traffic) from 0.5 x 103 particles cm-3 at 1 pm to 21 x 103 particles cm-3 at 3 pm; and (iii) factor 3 (aged road traffic) from 2.1 x 103 particles cm-3 at 2 pm to approx. 15 x 103 particles cm-3 at 5-6 pm."

Line 827: Quantify 'slightly affected'.

Reply: Done. See our reply to the previous point.

Line 843: 'Anomalously' implies that the measurements are flawed in some way. I do not believe that this is the case. Rephrase.

Reply: Done. "Anomalously" deleted.

Line 863: Clarify 'the fingerprint of London'.

Reply: Done. We have amended the sentences in the item: "An urban accumulation mode was found. This source presents a wide mode between 50-150 nm and accounts for around 10% of PNC. The wind directionality is consistent with the advection of air masses from London. It is more evident overnight due to the drop of the mixing layer top, the subsequent increase in air pollutants at ground level and the generation of nighttime secondary nitrate aerosols."

Figure 2: Check units of particle number concentrations – values are lower than typical ambient concentrations of 104 part/cm3 and are inconsistent with other figures. Reply: The unit is 103 particles/cm3. Corrected in Figure 2 and Figure SI4.

Figure 10: This could be moved to the SI as discussion of it is very brief. Supplementary material: This is not referenced in the main text. It is also lacking any descriptive text and is just a collection of figures. This does not provide the reader with accessible or helpful information.

Reply: Figure 10 (now Figure 9) is important for the discussion. Most of Section 3.4 refers to the plots shown in this figure. It is referenced in the text several times and
allows the reader to interpret the effects of the nucleation event upon the cluster and PMF results. On the contrary, we moved Figure 11 to SI (now Figure SI11), as its discussion is very brief.

All the tables and figures in the SI section are referenced in the main text.

REFEREE #2 The manuscript describes the results from a monitoring station close to LHR airport and uses two different analytical techniques to apportion or explain the particulate matter observed at the site. The data was collected during two intensives, nominally a warm and a cold season. The two analysis techniques are used to estimate the influence of the airport on the local particulate concentrations. It is clear the authors have analysed the data in detail and the use of the two techniques to interpret the data is good, as well as acknowledging the limitations. Overall, I think the work is suitable for publication. I have 3 main suggestions/questions and some detailed comments below.

Reply: We would like to thank anonymous referee #2 for his/her very useful suggestions and the general appraisal of the manuscript. Here are our answers. We have replied point-to-point to all the questions.

We have organised our replies by highlighting the referee comments in bold-italic font; our reply is in normal font.

We note that a second referee has asked us to shorten the manuscript sections related to the cluster analysis. We have therefore merged the discussion of the warm and cold seasons for the cluster analysis and we have summarised the main findings. We are confident that the whole discussion of results is now lighter, streamlined and easier to follow.

Main suggestions/questions: 1) I do not think the introduction is suitable as it does not provide a good background to the manuscript (see details below). In addition, neither the introduction or the abstract mention the use of the regional nucleation event, which is really important for understanding a potential limitation in the two analysis techniques

ACPD
Reply: In response to the points raised by both referees, we have extensively improved the "Introduction" section. In particular, we have (i) improved the literature review by adding a number of up-to-date references, (ii) improved the description of the objectives of the study, (iii) added a discussion over the novel contribution of this study with respect to the available literature, and (iv) moved some references and text from the "results and discussion" to the "introduction".

We are confident that the current version of the "Introduction" provides all the information necessary to summarise the state-of-the-art of the research on the PNC pollution close to airports. We also believe that the improved version of the "Introduction" is able to place our study in the context of others and well defines its novel contribution to the science.

We have also added some sentences in the abstract and in the introduction presenting and describing the analysis of the strong nucleation event which occurred during the sampling campaign.

2) I think a lot of the NOx and O3 details and graphs can be removed. The authors acknowledge that NOx has been studied in detail at the site before and the contributions from LHR to the site have been published. The paper itself is about the particulates, not air pollution in general. This would streamline the manuscript.

Reply: Done. The whole subsection 3.1 was shortened and details on the NOx and O3 were removed as well as Figure 2a (completely moved to SI). We have also modified the former Figure 2b (now Figure 2): now this figure shows only the variables discussed in the main text and important for the following discussion. The diurnal profiles for all the variables (former Figure 2b) has been moved to SI (new Figure SI4).

3) What version of AIM was used to analyse the data? If it was version 10, then no action is required. If it was version 9 however, then some work is needed. There is a significant difference between V9 and V10 in the diffusional correction algorithms. V9 over estimates the size of the correction needed and can lead to an over estimation
of the nanoparticles, especially below 50nm. As the authors are dividing the particles into Aikten and Ultra fine modes (and the characteristics of these modes change with season), there is a need to check as it could lead to artefacts in the clustering.

Reply: We used the version 8.3, whose outputs we confirmed to be identical to the version 9. Despite the differences in the algorithm used for correcting the diffusion losses between v.10 and previous versions, TSI UK have confirmed that since we used a TSI 3080 electrostatic classifier with a long differential mobility analyser (TSI 3081), which were released with AIM v.9 we were correct to use this version of the software. In addition, since previous studies in London are also based on data extracted with those previous versions of AIM, our results are easily comparable with some of the most relevant literature.

Minor question: Has the Aethelometer data been corrected according to Collaud Coen et al., 2010? I note the reported protocol as defined in Petzold et al., 2015. If not, it needs correcting especially if the authors want to draw BB aerosol information from it.

Reply: No. Data are corrected using the WUAQL AethDataMasher V7.1 to perform data validation and correct data for non-linear loading effects. We cannot apply any further correction to BC data as no reference instrument was run. Anyway, we would like to point out that this study focuses on the PNSDs and black carbon data are only used as ancillary variables to be related with the particle sources. In addition, we would like to report that a large proportion of the black carbon data in the literature are derived from aethalometers with no more than a loading correction.

A sentence was added to the text: "eBC was derived from the absorbance at 880 nm wavelength; raw data were post-processed with the Washington University Air Quality Lab AethDataMasher V7.1 to perform data validation and correct data for non-linear loading effects (Virkkula et al., 2007; Turner et al., 2007)."

Details: I question the relevance of some of the introduction to the manuscript. Lines 75 to 83 detail the pros and cons of a third runway, which I do not think adds to the
manuscript as the study is about the impacts of Heathrow. It is fine to say the airport is planning to increase capacity, but as to why is not for this manuscript. In line 82, the use of the term 'Despite this' could be misinterpreted as questioning the UK government's decision.

Reply: A large number of sentences from this paragraph have been removed, as also recommended by the comments of another referee. As a consequence "despite this" was also deleted. The sentence on the future planned expansion of LHR has now a governmental reference (UK Dept. for Transport).

Lines 85 – 104 Again, I think the context of this paragraph is not appropriate for the manuscript. The manuscript is about characterising the particulates close to Heathrow and not about questioning EU air quality standards as there is nothing in the manuscript that takes this data and compares mortality or morbidity predicted from these measurements with those based on the EU standard method, for example. This section should be that despite the UK meeting the PM standard, it has been show UFP have health issues etc and because there is no network of measurements, campaigns such as this are key to assessing the sources and potential impacts of airports on the UFP. The last sentence should be removed from this section. As the authors point out (line 98), there is limited knowledge in this area, so what should be regulated and how? EU has standards on particulates from cars. New aircraft will soon be regulated on particulate number and mass. EU law requires restricting exposure to nano-particles in the work place. The UK has the clean air act. So there are policies in place, but more info in needed before guidelines and strategies can be applied.

Reply: The section has been comprehensively revised and shortened according to the referees' comments.

Section 2.1 – study area and dates. With the exception of the first sentence, this section should be in the introduction. It is background and context material.

Reply: Done.
Lines 150-151, no need to repeat the dates here as they were just in the previous section.

Reply: Done.

Line 151-152, data from the site is QA etc. Which data? The authors or the data from the AURN site?

Reply: AURN site. This sentence was removed. We refer to the QA of DEFRA data later in Section 2.1.

Lines 288 – 303, this reads like an introduction and is better suited in the introduction.

Reply: Done. Moved to "Introduction", as also recommended by Referee #1.

Section starting at line 309. This is not an acceptable way to present data and I don't see what ranking them in order shows. This also seems to repeat a lot of information already contained within figure 2a. Suggest removing most of that section and simply referring to fig 2a, M&H 2015 and that the data is representative.

Reply: Done. The "ranking" was removed. The whole section was also shortened and improved. Some minor typing errors were also found and corrected.

Line 318, Suggest changing 'Since' to 'Analysis of the data showed it was not: : :..' and provide a reference for the Kruskal-Wallis analysis.

Reply: Done: sentence modified and reference added.

Line 324 and figure 3. Firstly, if the PNSD are shifting towards coarser modes, how can the PVSD be almost constant? The author's choice of wording is contradictory. Secondly, in the summer it appears the mode during the day could be anywhere between 20 and 50nm and during the night it is around 35nm. Is the statement backed up by fitting a double log-normal to the data? Thirdly, why aren't the percentiles for the PVSD presented if they are shown for the PNSD? Finally, why are the medians presented and not the means, which is more common? Does the later analysis use the means or
medians? If the former, figure 3 should show means.

Reply: The discussion of the PNSDs and PNVDs has been improved according to the referee's comments. In addition, the mode ranges presented in the text have been corrected, as highlighted by the referee. We have also fitted the size spectra with lognormal curves, by using 2 to 4 curves. However, we do not believe that this fitting analysis should be included in the manuscript and only served as a preliminarily investigation of the PNSDs.

This Figure (now Figure 1) was originally planned to be presented as a boxplot (where IQR (25th-75th percentile range) and median are shown). This is the reason for having selected these statistics. However, we recognize that it is also useful to provide the average distributions (more common, but potentially affected by outliers). Therefore, we have added them for PNSD and PVSD as dotted lines.

We only showed the IQRs for PNSD because these are the data measured by the SMPS. PNVD are subsequently derived by the simple assumption that all particles are perfectly spherical (which -we know- isn't true). Therefore, the modeling of PNVD is affected by a large uncertainty and in this study is only used for descriptive purposes (not for quantitative analysis). Consequently, we believe that it makes no sense to go into a deep analysis of the PNVD. In addition, we want to keep the figures as simple as possible and the addition of other curves to the plot can mislead the reader and distract from what is really important here, i.e. the PNSDs.

Finally, the PVSD are almost constant because the second mode of the PNSD does not change during the day/night. The text has been improved accordingly.

The amended paragraph: "The average PNSDs are shown in Figure 2 as well as their median distributions and interquartile ranges. Spectra are categorised by time of day (7am-7pm and 7pm- 7am local time). In addition, the particle volume size distributions (PVSDs) are also provided. Results for the warm season show that the average day-time PNSD is dominated by a main peak in the nucleation range (extending below 14

ACPD
nm) and a second mode in the Aitken range (between 30 and 50 nm). The nocturnal spectrum is characterised by a drop of the nucleation mode to concentration values similar to the Aitken peak (mode around 35 nm). During the cold season, the average diurnal and nocturnal PNSDs present a main peak at 15-25 nm and a second mode at 70-100 nm. In summary, both seasons show reductions of the finest modes during nighttime, while the second mode is almost constant throughout the day. As a consequence, the modal structure of PNVDs is also almost constant throughout the day.".

Line 330, meteorology plays a role.

Reply: A sentence was added to the text.

Line 345, constant to 11pm (same value pre noon in summer)

Reply: The sentence has been amended.

Line 349, the accumulation mode peaks at around 10pm, which is inconsistent with the statement that it follows traffic which peaks at 5-6pm, for the M4 at least.

Reply: An improved explanation was added. We believe that the dynamics of the mixing layer may play a role in extending the evening peaks up to the night. The amended text: "Generally, the evening peaks start around 6 pm, which is consistent with the peak of traffic (Figure SI5) but they extend late in the evening and night probably because the drop of the mixing layer and the consequent concentration of pollutants close to ground level.".

Line 351, I don't think the use of the word intermediate is appropriate. It is either traffic or aircraft or mixed, perhaps. Furthermore, this is interesting. Why is there a difference in winter and summer? If Version 9 of AIM was used, could this change your results if you try version 10?

Reply: "Intermediate" has been replaced with "mixed". We do not attempt a further explanation of the diurnal profiles of the Aitken nuclei in this section. We limit our dis-

**ACPD**
cussion to presentation of the results. We believe that this "mixed" behavior is related to the influence of different sources and is also complicated by the different meteorological conditions between the two seasons.

The amended sentences: "Aitken nuclei exhibit a mixed behaviour between nucleation and accumulation particles (Figure 2): two different patterns can be found, which are more consistent with road traffic in summer and with aircraft traffic in winter.".

We have not investigated if this pattern is also detected when using AIM10 to extract the data. We believe that the manuscript should only focus on data extracted with AIM V9 (see reply to the referee's main point 3).

Lines 355 - 381. I do not think this section needs to be in the manuscript. The title of the manuscript is about the sources of the particles near an airport. The NO2 is a) not showing any directionality and b) has been extensively studies already. The same is true for figures 2 a and b, the NOx data can be removed from there to streamline the paper.

Reply: We have removed the paragraphs on the NOx and O3. The former Figure 2a has been moved to SI. The former Figure 2b (now Figure 2) was modified to show the most important daily patterns in view of the interpretation of the results. The diurnal patterns for all variables are now provided in the SI.

Line 424 – I think cluster 2 can be bi modal as well, it certainly shows in the percentiles.

Reply: The whole section on the cluster analysis has been modified. Now, we jointly discuss Cluster 2 and 3 for the warm season and the Cluster 2 for the cold season. All these clusters have bimodal structures and are related to the same source.

Line 429 - suggest changing to 'the POTENTIAL role: : ...'

Reply: Done.

Line 445 - again, is it mono-modal or bi-modal? The small bump at 14nm is in the
median and percentiles.

Reply: Done. The whole section on the cluster analysis has been modified. Now, we jointly discuss Cluster 5 for the warm season and the Cluster 3 for the cold season. All these clusters have a main peak at 20-35 nm. We added discussion of the second possible peak of Cluster 5 within brackets.

Line 457 & figure 4. The hourly count for cluster 4 is very noisy. Where is the morning peak? The one just after 6am? I also disagree it is the mirror image of clusters 2 & 3 counts. It shows a broadly opposite trend. Suggest rewording.

Reply: This was unimportant to the main purpose of the paper and has been deleted. Cluster 1 & 5 in the cold season – It seems a little odd that the count profile of cluster 1 matches more the profile of LHR than cluster 5, yet the suggestion is that cluster 1 is an aged LHR aerosol, and cluster 5 is fresh because short transit times (high wind speed). Surely a fresh emission will match more the LHR profile, while the process of aging will remove or diminish the effects of source? I don't think the author's conjecture is correct for cluster 5 being fresh emissions.

Reply: We have moved our discussion on these clusters from this section to Section 3.3 "Comparison of k-means and PMF". This latter analysis has allowed a better interpretation of the Clusters 1 and 5 (cold season). Essentially, Cluster 5 presents high contributions of the "airport" source of PMF and, therefore, may represent the spectra shaped largely by the airport emissions. Cluster 1 also has high contributions for the PMF factor associated with road traffic. Both are mixed spectra.

Cluster 3 cold season – can you back up your BB conjecture with the Aeth delta C data averaged over the same periods? If not, you should remove it.

Reply: There is not a significant increase of delta-C. We have therefore removed this "unsupported" interpretation. We have added a revised interpretation for this result: the winter mixing layer effects.
The amended text: "However, the diurnal pattern in winter also presents a high number of counts at 3-5 am, i.e. not directly compatible with rush hours. A possible explanation involves the stronger effect of the winter mixing layer dynamics on the air quality due to the presence of more frequent low level thermal inversions, which may build up the pollutants at ground-level especially overnight. This may increase the signal of the less intense, but still significant, nighttime traffic emissions present in the study area."

Line 509 replace hump with mode.

Reply: Done.

Line 581 – The contribution to the NO2 levels at LHR are quoted here as 25-30%, but in the previous section () is was found to be 15-17% from another study. Was this study different to previous ones?

Reply: We have removed the sentences in line 367. We recognise that this discussion is misleading. The data refer to two different studies carried out with different apportionment methods and using data from different periods (2002-2004 and 2015-2012). In addition, we have now removed the section discussing the NOx data (see referee's main point 2).

A review of the results available in the literature is presented in the "Introduction": "For example, Carslaw et al. (2006) estimated that airport operations in 2001/4 accounted for ~27% of the annual mean NOx and NO2 at the airfield boundary and less than 15% (

Line 694 – 'to' missing from second sentence.

Reply: Amended: "to" added.

Line 711 - 3% is not modest, it is minimal.

Reply: Amended, "modest" substituted with "minimal".

Line 848 – suggest removing Anomalously . Being downwind of an airport is expected to lead to higher loadings.

Reply: Done.

---

## Author Response (AR2)

| 1
2
3
4
5 | Journal: ACP
MS No.: acp-2017-150
Title: Sources of Submicrometre Particles Near a Major International Airport
Author(s): Mauro Masiol et al. |
|-----------------------|--------------------------------------------------------------------------------------------------------------------------------------------------------|
| 6
7                | RESPONSE TO EDITOR                                                                                                                                     |
| 8                     | Co-Editor Decision: Reconsider after minor revisions (Editor review)                                                                                   |
| 9
10               | (27 Aug 2017) by Andreas Petzold                                                                                                                       |
| 11
12
13        | Comments to the Author:
Dear Roy Harrison                                                                                                           |
| 14
15              | The manuscript requires two minor revisions before being accepted for ACP:                                                                             |
| 16
17              | 1. The abstract is overly long at >350 words. I suggest cutting this down to ~250.                                                                     |
| 18                    | RESPONSE: This has been reduced in length by 87 words.                                                                                          |
| 19                    |                                                                                                                                                        |
| 20                    | 2. Line 198: Define eBC.                                                                                                                               |
| 21                    | RESPONSE: This is now defined by a reference.                                                                                                   |

[revised manuscript text omitted]